# White plague among the "forgotten people" from the Barbaricum of the Carpathian Basin – Cases with tuberculosis from the Sarmatian-period (3rd–4th centuries CE) archaeological site of Hódmezővásárhely–Kenyere-ér, Bereczki-tanya (Hungary)

Olga Spekker[1,2,3]*, Attila Kiss P.[4], Luca Kis[3,5], Kitty Király[3,6], Sándor Varga[6], Antónia Marcsik[3], Oszkár Schütz[5,7], Tibor Török[1,5,7], David R. Hunt[8], Balázs Tihanyi[3,5]

1 Ancient and Modern Human Genomics Competence Centre, University of Szeged, Szeged, Hungary, 2 Institute of Archaeological Sciences, Eötvös Loránd University, Budapest, Hungary, 3 Department of Biological Anthropology, University of Szeged, Szeged, Hungary, 4 Department of Early Hungarian and Migration Period Archaeology, Pázmány Péter Catholic University, Budapest, Hungary, 5 Department of Archaeogenetics, Institute of Hungarian Research, Budapest, Hungary, 6 Department of Archaeology, Móra Ferenc Museum, Szeged, Hungary, 7 Department of Genetics, University of Szeged, Szeged, Hungary, 8 Virginia Office of the Chief Medical Examiner, Northern District, Manassas, Virginia, United States of America

* olga.spekker@gmail.com

## Abstract

Tuberculosis (TB) is a bacterial infection that is well-known in the palaeopathological record because it can affect the skeleton and consequently leaves readily identifiable macroscopic alterations. Palaeopathological case studies provide invaluable information about the spatio-temporal distribution of TB in the past. This is true for those archaeological periods and geographical regions from when and where no or very few TB cases have been published until now–as in the Sarmatian period (1st–5th centuries CE) in the Barbaricum of the Carpathian Basin. The aim of our paper is to discuss five newly discovered TB cases (**HK199**, **HK201**, **HK225**, **HK253**, and **HK309**) from the Sarmatian-period archaeological site of Hódmezővásárhely–Kenyere-ér, Bereczki-tanya (Csongrád-Csanád county, Hungary). Detailed macromorphological evaluation of the skeletons focused on the detection of bony changes likely associated with different forms of TB. In all five cases, the presence of endocranial alterations (especially TB-specific granular impressions) suggests that these individuals suffered from TB meningitis. Furthermore, the skeletal lesions observed in the spine and both hip joints of **HK225** indicate that this juvenile also had multifocal osteoarticular TB. Thanks to the discovery of **HK199**, **HK201**, **HK225**, **HK253**, and **HK309**, the number of TB cases known from the Sarmatian-period Carpathian Basin doubled, implying that the disease was likely more frequent in the Barbaricum than previously thought. Without the application of granular impressions, the diagnosis of TB could not have been established in these five cases. Thus, the identification of TB in these individuals highlights the importance of

**Data Availability Statement:** All relevant data are within the paper and its Supporting Information files.

**Funding:** This work was funded by the Cooperative Doctoral Programme for Doctoral Scholarships 2020 (grant agreement no. 1020404) of the Hungarian Ministry of Innovation and Technology to LK, by the University of Szeged Open Access Fund (grant agreement no. 6325) to OS, and by the Competence Centre of the Life Sciences Cluster of the Centre of Excellence for Inter-disciplinary Research, Development and Innovation of the University of Szeged to TT. This project received funding also from the European Research Council (ERC) under the European Union's Horizon 2020 research and innovation programme (grant agreement no. 856453 ERC-2019-SyG). The funders had no role in study design, data collection and analysis, decision to publish, or preparation of the manuscript.

**Competing interests:** The authors have declared that no competing interests exist.

diagnostics development, especially the refinement of diagnostic criteria. Based on the above, the systematic macromorphological (re-)evaluation of osteoarchaeological series from the Sarmatian-period Carpathian Basin would be advantageous to provide a more accurate picture of how TB may have impacted the ancestral human communities of the Barbaricum.

## Introduction

Palaeopathology (the scientific study of ancient diseases in skeletonised and mummified human remains) is a key contributor to the field of bioarchaeology, exploring the disease experience of individuals in prehistoric and historic times [1–5]. Initially, palaeopathology focuses on the individual, and its main objective is to determine where and when a disease was present in the past [4–7]. Establishing a definitive (individual) diagnosis is based on the recognition of pathological bony changes and their distribution pattern in the skeleton, and the elimination of alternative aetiologies through differential diagnosis [4–7]. It is easily realised that in palaeopathological practice, it is challenging to arrive at a definitive diagnosis because: 1) In most cases, only bone remains are available for evaluation; thus, only those diseases which directly or indirectly affect the skeleton can be identified [8, 9]. 2) Even if a disease can manifest on the bones and it was present in an individual at the time of death, it may not have been afflicting the individual for a sufficiently long time to have bony changes to develop [4, 10]. 3) Even if the disease has manifested on their bones, the ability to notice alterations during evaluation is highly dependent on the completeness and preservation of the individual's skeleton [10, 11]. And 4) even if bony changes can be detected on the skeleton, it is quite difficult to establish a definitive diagnosis relying solely on the observable macroscopic alterations [1, 10, 12]. This is because bone can react in a limited number of ways, and numerous diseases (or even taphonomic processes) can cause the same or strikingly similar skeletal lesions [1, 10–13]. Because of these limitations, only the specific distribution pattern of bony changes in the skeleton can provide a definitive diagnosis [1, 10–13].

One of the main fields of interest in palaeopathology is infections [3, 12, 14]. Tuberculosis (TB) has been afflicting humankind for millennia and can be considered responsible for more deaths in history than any other communicable disease [3, 5, 12, 14–19]. TB is well-known in the palaeopathological record because it can directly or indirectly affect the skeleton, leaving readily identifiable macroscopic alterations [3–5, 7, 12, 14–21]. Palaeopathological case studies provide invaluable information about the spatio-temporal distribution of TB in the past [4, 5, 7]. This is true for those archaeological periods and geographical regions from when and where no or very few TB cases have been published until now–as in the Sarmatian period (1st–5th centuries CE) in the Barbaricum of the Carpathian Basin. Only three TB cases have been reported from the 4th–5th-century-CE archaeological site of Apátfalva–Nagyút dűlő by Marcsik & Kujáni [22] (detailed information regarding the Sarmatian period in the Barbaricum of the Carpathian Basin can be found in S1 Text).

The anthropological investigations of skeletal remains from the Sarmatian period of the Carpathian Basin were begun in the first third of the 20th century CE [23, 24], and numerous researchers [25–34] have focused on the biological reconstruction of this period and peoples. Despite a number of studies that have been written, little is known about these populations, especially since the majority of the excavated skeletons are incomplete and poorly preserved. The lack of well-preserved remains likely contributed to the low number of reported TB cases

[35]. The aim of our paper is to demonstrate and discuss in detail five cases (**HK199**, **HK201**, **HK225**, **HK253**, and **HK309**) from the Sarmatian-period (3rd–4th centuries CE) archaeological site of Hódmezővásárhely–Kenyere-ér, Bereczki-tanya (Csongrád-Csanád county, Hungary), whose skeleton exhibited bony changes attributable to TB. Furthermore, we will compare these newly discovered TB cases with the three from the Apátfalva–Nagyút dűlő archaeological site, focusing on 1) the age at death and sex of the affected individuals, 2) the presented disease form, and 3) the macroscopic characteristics (e.g., location and extent) of the observed alterations.

## Materials and methods

### Materials

The Hódmezővásárhely–Kenyere-ér, Bereczki-tanya archaeological site (site ID: 56669) is geographically located north of the present-day town of Hódmezővásárhely (Csongrád-Csanád county, Hungary), at the southern side of a bigger bend of the Kenyere Creek (Fig 1A) [36]. Due to construction work of the Hódmezővásárhely bypass section of main road #47, a preventive excavation between 2015 and 2017 was undertaken at the site by staff from the Móra Ferenc Museum (Szeged, Csongrád-Csanád county, Hungary) [36]. During the two-year-long excavation period, 330 archaeological objects were unearthed from the large excavation area (42,943 m$^2$) (Fig 1B). Most of these objects derive from the Roman Age, including the remains of a Sarmatian-period settlement where pits were the most common findings [36]. Besides the settlement objects, three Sarmatian-period cemeteries were also discovered at the site [36]. It should be noted that all these cemeteries can be considered only partially excavated since they extend further beyond the limits of the excavation [36].

The three cemeteries are separated from each other as well as from the settlement and are situated on small ground elevations [36]. From the first cemetery (Fig 1C), nine south-north oriented inhumation graves were exposed, the majority of the grave pits were rectangular in shape with slightly rounded corners [36]. The burials had a scattered distribution and were clustered in smaller groups [36]. Two of these burials (object nos. 195 and 204) were surrounded by interrupted ditches, but there could have been ditches around another grave as well (object no. 201) [36]. The second cemetery was located south of the first one (Fig 1D) and was comprised of four rows of uniformly oriented (south-north), mostly rectangular-shaped inhumation graves [36]. Among the 18 uncovered burials, seven (object nos. 253, 255, 256, 257, 309, 312, and 313) were surrounded by interrupted ditches [36]. There could have been ditches around another grave as well (object no. 244) [36]. In contrast to the first cemetery, there were signs of contemporary grave robbery in a number of burials–for instance, the seven graves with ditches [36]. The third cemetery was situated about 150 m west of the second one and only three graves were unearthed from it [36]. One of them (object no. 316) was surrounded by interrupted ditches and had been robbed [36]. There were grave goods buried along with the deceased in the three Sarmatian-period cemeteries–jewellery (e.g., earrings, bracelets, and fibulae), beads, iron objects (e.g., knives, buckles, and fibulae), Roman-Age coins, and ceramic vessels were the most common artefacts [36]. Dating by the associated grave goods indicates that the three cemeteries were in use in the 3rd–4th centuries CE [36]. From the Sarmatian-period archaeological site of Hódmezővásárhely–Kenyere-ér, Bereczki-tanya, a total of 28 human skeletons were unearthed– 27 were uncovered from the three cemeteries, and one was discovered in a pit (object no. 225) found among the Sarmatian-period settlement remains.

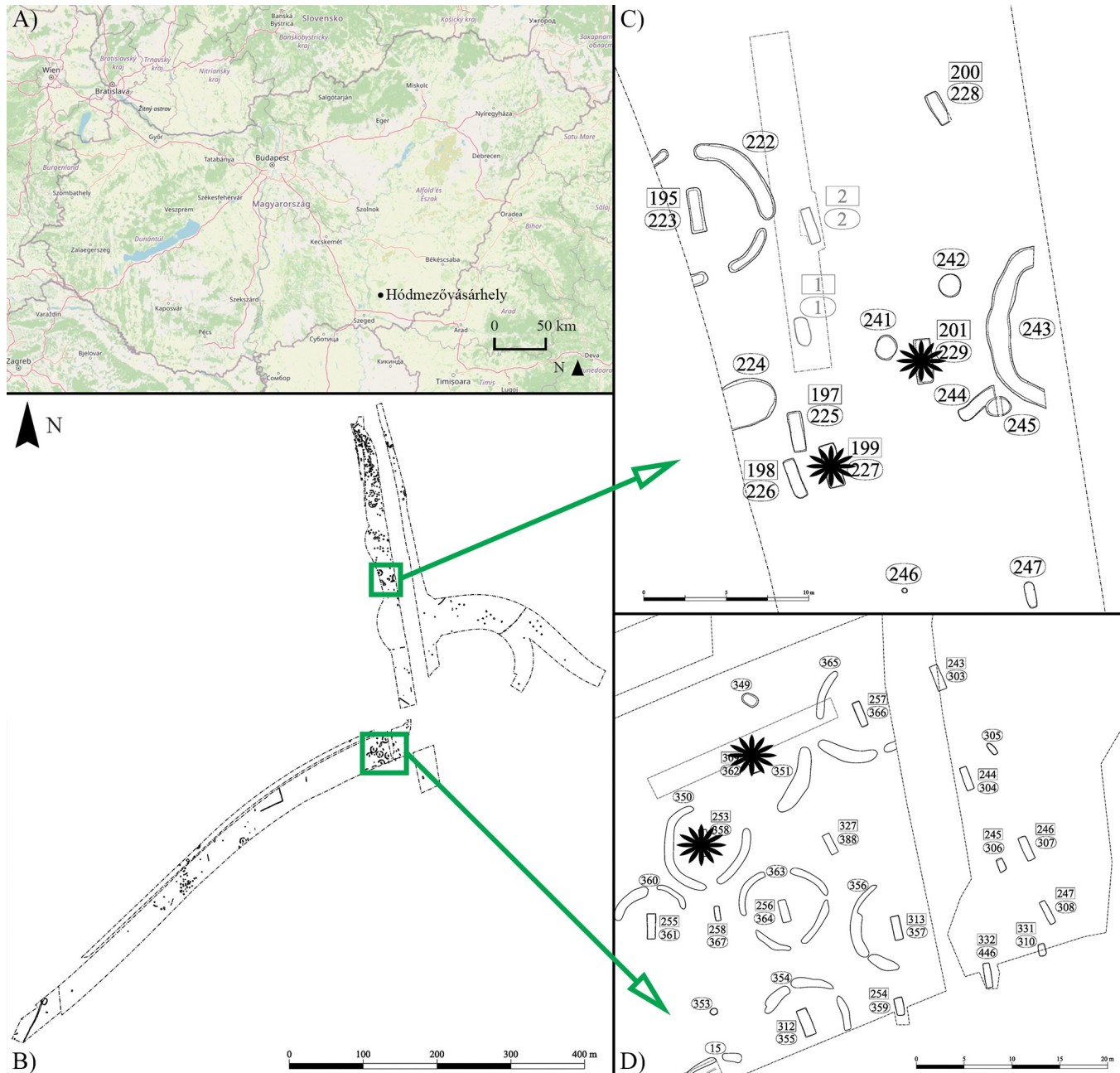

**Fig 1.** A) Map of Hungary showing the location of the Hódmezővásárhely–Kenyere-ér, Bereczki-tanya archaeological site; B) Plan drawing of the Sarmatian-period archaeological site of Hódmezővásárhely–Kenyere-ér, Bereczki-tanya with the location of the first and second cemeteries (green rectangles); C) Plan drawing of the first Sarmatian-period cemetery from the Hódmezővásárhely–Kenyere-ér, Bereczki-tanya archaeological site with the location of the graves of HK199 and HK201 (black stars); and D) Plan drawing of the second Sarmatian-period cemetery from the Hódmezővásárhely–Kenyere-ér, Bereczki-tanya archaeological site with the location of the graves of HK253 and HK309 (black stars). (Fig 1A contains information from OpenStreetMap and OpenStreetMap Foundation, which is made available under the Open Database License).

## Methods

No permits were required for the described study, which complied with all relevant regulations. The 28 human skeletons uncovered from the Sarmatian-period (3rd–4th centuries CE)

archaeological site of Hódmezővásárhely–Kenyere-ér, Bereczki-tanya are currently housed at the Móra Ferenc Museum (Szeged, Hungary). During the detailed anthropological investigation, all bone remains were macroscopically inspected with the naked eye. Age at death [37–45] was estimated and sex [46–48] was determined applying standard macromorphological methods of bioarchaeology. Completeness of the skeleton was ranked: 1) almost complete–more than 80% of the skeleton is extant; 2) relatively complete– 50–80% of the skeleton is extant; or 3) very incomplete–less than 50% of the skeleton is extant. Preservation of the skeleton was ranked: 1) well-preserved–there is minimal overall *post-mortem* damage of the bone surfaces; 2) fairly preserved–there is moderate overall *post-mortem* damage of the bone surfaces; or 3) poorly preserved–there is significant overall *post-mortem* damage of the bone surfaces.

After these analyses, a detailed palaeopathological evaluation was performed on all 28 human skeletons. It must be noted that the missing and taphonomically damaged bones of the examined individuals substantially hindered these observations. The registration of pathological alterations indicative of TB was based on the following:

1. Skeletal lesions suggestive of osteoarticular TB were identified following the guidelines of Aufderheide & Rodríguez-Martín [1], Baker [49], Mariotti and co-workers [50], Ortner [12], and Roberts & Buikstra [20];

2. Bony changes indicative of pulmonary TB/TB pleurisy were recorded considering the descriptions of Assis and colleagues [51], Kelley & Micozzi [52], Matos & Santos [53], Roberts and co-workers [54], Rothschild & Rothschild [55, 56], Santos & Roberts [57, 58], and Winland and colleagues [59]; and

3. Alterations suggestive of TB meningitis were identified following the recommendations of Hershkovitz and co-workers [60], Schultz [61–64], Schultz & Schmidt-Schultz [65], and Spekker and colleagues [66–69].

In the current study, five individuals (**HK199**, **HK201**, **HK225**, **HK253**, and **HK309**) were selected for detailed discussion since they exhibited skeletal lesions indicative of certain forms of TB:

- **HK199 (object no./stratigraphic no.: 199/227):** a younger to middle-aged adult (c. 20–39/ 49 years old [39, 43, 45]) female [46–47]. This skeleton is relatively complete and fairly preserved; the remains were unearthed from the first cemetery (Fig 1C);

- **HK201 (object no./stratigraphic no.: 201/229):** a middle-aged to older adult (c. 40–x years old [39, 43]) individual of indetermined sex [46]. This skeleton is very incomplete and poorly preserved; the remains were uncovered from the first cemetery (Fig 1C);

- **HK225 (object no./stratigraphic no.: 225/302):** a juvenile (c. 14–16 years old [40]) individual of indetermined sex [46, 47]. This skeleton is relatively complete and well-preserved; the remains were discovered in a pit from the Sarmatian-period settlement;

- **HK253 (object no./stratigraphic no.: 253/358):** a younger adult (c. 20–29 years old [38, 39, 43–45]) female [46, 47]. This skeleton is relatively complete and fairly preserved; the remains were unearthed from the second cemetery (Fig 1D); and

- **HK309 (object no./stratigraphic no.: 309/362):** a middle-aged to older adult (c. 50–x years old [37, 39, 41, 43–45]) male [46, 47]. This skeleton is relatively complete and fairly preserved; the remains were uncovered from the second cemetery (Fig 1D).

### Ethics statement

Specimen numbers: **HK199** (object no./stratigraphic no.: 199/227), **HK201** (object no./stratigraphic no.: 201/229), **HK225** (object no./stratigraphic no.: 225/302), **HK253** (object no./stratigraphic no.: 253/358), and **HK309** (object no./stratigraphic no.: 309/362).

The five skeletons from the Hódmezővásárhely–Kenyere-ér, Bereczki-tanya archaeological site (site ID: 56669) evaluated in the described study are housed in the Móra Ferenc Museum, in Szeged, Hungary. Access to the specimens was granted by the Department of Archaeology of the Móra Ferenc Museum (Roosevelt tér 1–3, H-6720 Szeged, Hungary).

No permits were required for the described study, which complied with all relevant regulations. The research has been conducted in an ethically responsible manner–the bone remains of **HK199**, **HK201**, **HK225**, **HK253**, and **HK309** have been examined with dignity and respect.

### Results

#### HK199 (younger to middle-aged adult female)

During the macroscopic evaluation of the skeleton of **HK199**, the inner surface of the cranium revealed three alteration types that can be attributed to TB. Although the taphonomic damage precluded the definitive observation of some regions of the endocranial surface (e.g., squamous part of the occipital bone and left temporal bone), granular impressions (GIs) were detected at the junctions of the squamous and orbital parts of the frontal bone (Fig 2) and on the squamous part of the right temporal bone. Furthermore, abnormal blood vessel impressions (ABVIs) and periosteal appositions (PAs) were noted on both parietal bones and the squamous part of the frontal bone (Fig 3B and 3C) and occipital bone (Fig 3A). In the sulci of the superior sagittal sinus and the left transverse sinus, there were patches where these two lesion types occurred not separately but accompanying each other.

#### HK201 (middle-aged to older adult individual)

The endocranial surface of **HK201** exhibited skeletal lesions that are indicative of TB. Both parietal bones (Fig 4A, 4B, 4D and 4E) and the left temporal bone (Fig 4C) displayed GIs–close

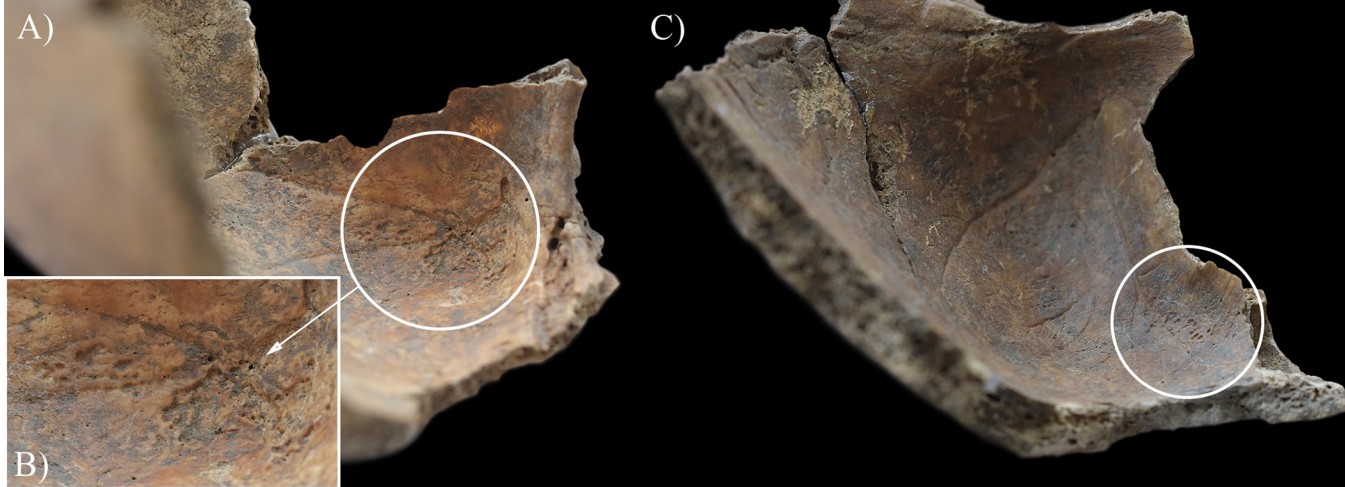

**Fig 2.** Endocranial granular impressions (white circles) at the junctions of the squamous and orbital parts of the frontal bone of HK199: A) left side, B) close-up of the left side, and C) right side.

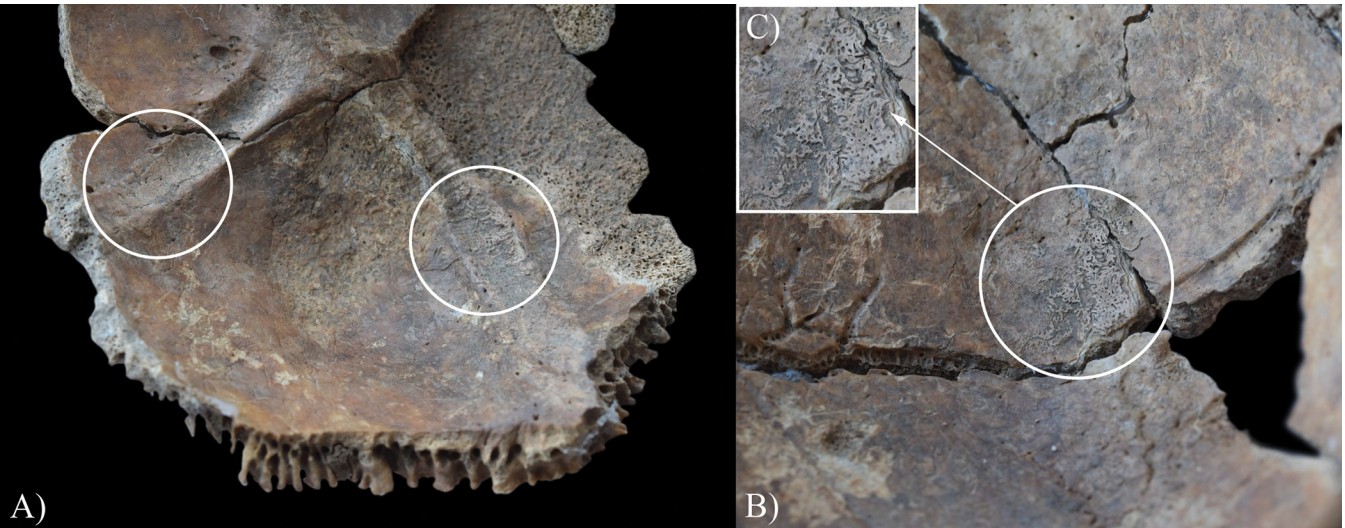

**Fig 3.** Endocranial abnormal blood vessel impressions and periosteal appositions (white circles) on the squamous part of the A) occipital bone (in the sulci of the superior sagittal sinus and the left transverse sinus) and B) frontal bone (in the sulcus of the superior sagittal sinus) of HK199. C) Close-up of the abnormal blood vessel impressions and periosteal appositions on the squamous part of the frontal bone of HK199.

to the squamous suture and on the squamous part, respectively. It should be noted that some of the predilection sites of GIs (e.g., the squamous part of the occipital bone and the right temporal bone, and the greater wings of the sphenoid bone) were severely damaged or missing *post-mortem*, precluding the definitive observation of their endocranial surface.

## HK225 (juvenile individual)

In contrast to the other cases, in the skeleton of **HK225**, not only the endocranial surface but the ectocranial surface, the spine, and the pelvic area also showed bony changes that are associated with TB. The inner surface of the frontal bone (at the junctions of the squamous and orbital parts) (Fig 5C and 5D), both temporal bones (squamous part) (Fig 5A and 5B), the occipital bone (squamous part) (Fig 6A–6E), and a fragment of the sphenoid bone (greater

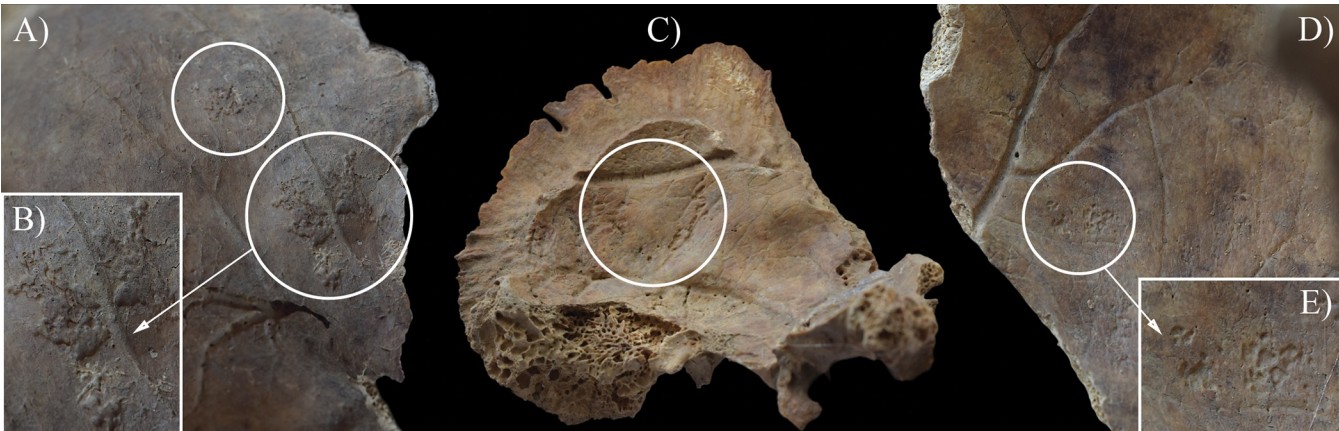

**Fig 4.** Granular impressions (white circles) on the endocranial surface of HK201: A) right parietal bone (close to the squamous suture), B) close-up of the right parietal bone, C) left temporal bone (squamous part), D) left parietal bone (close to the squamous suture), and E) close-up of the left parietal bone.

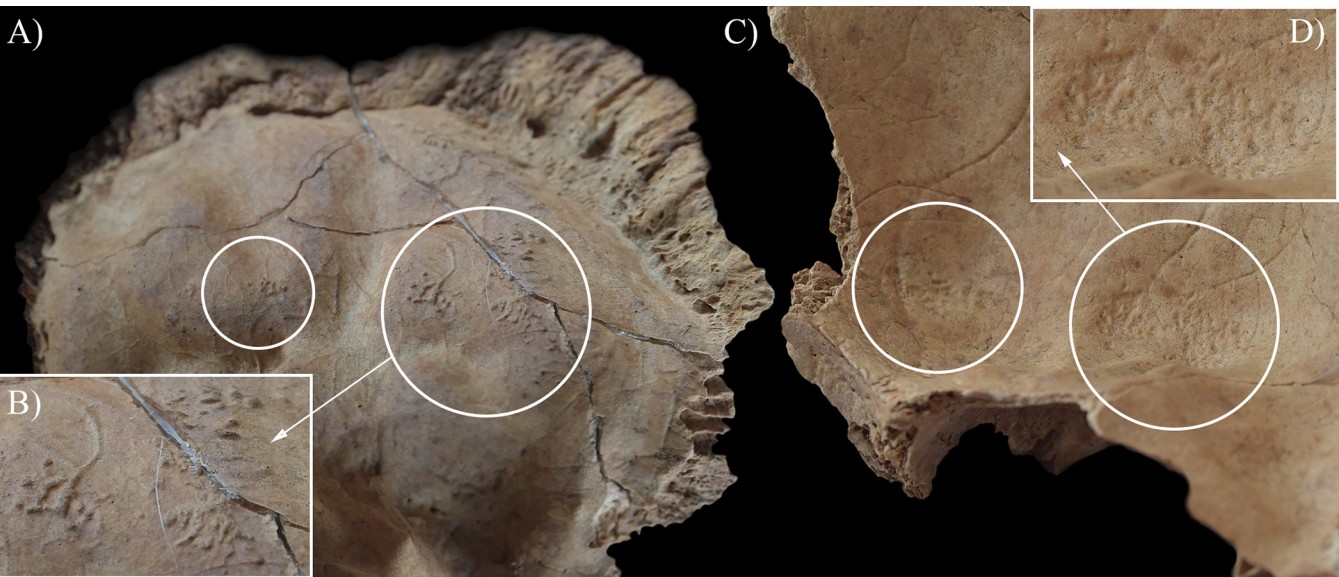

**Fig 5.** Granular impressions (white circles) on the endocranial surface of HK225: A) right temporal bone (squamous part), B) close-up of the right temporal bone, C) left side of the frontal bone (the junction of the squamous and orbital parts), and D) close-up of the left side of the frontal bone.

wing) exhibited GIs. Moreover, ABVIs and PAs were detected on the frontal bone (along the sulcus of the superior sagittal sinus) (Fig 7B and 7C), both parietal bones (along the sulcus of the superior sagittal sinus) (Fig 7A), and the occipital bone (along the sulci of the superior sagittal sinus and both transverse sinuses) (Fig 6A, 6C–6E). The squamous part of the occipital bone presented abnormally pronounced digital impressions (APDIs) (slight stage) (Fig 6E). Besides the endocranial alterations, the ectocranial surface of the occipital bone displayed pitting and slight cortical remodelling on both lateral parts (around the occipital condyles) (Fig 8A); the pathological process spared the articular surfaces. In the spine, the most severe pathological bony changes were observed in the T2 and L5 vertebral bodies. The second thoracic vertebral body was almost completely destroyed with consequential collapse into a slightly wedge-shape (Fig 9). The fifth lumbar vertebral body showed an oval-shaped, well-circumscribed osteolytic lesion at the upper subchondral region (Fig 8C). In the lower cervical (C5–7), thoracic (T1 and T3–12), and lumbar (L1–5) regions (Figs 8B and 10), the anterior and/or lateral aspects of the vertebral bodies revealed signs of hypervascularisation (in the form of multiple, circumferential, smooth-walled, resorptive pits (i.e., enlarged vascular foramina), which were occasionally connected by horizontal, superficial abnormal blood vessel impressions). Similar alterations were observed on the pelvic surface of the sacrum (S1–4) (Fig 11A). Both sacral wings presented surface pitting and reactive new bone formations (Fig 11). Surface pitting was observed in the lower thoracic and lumbar regions on the anterior aspect of the vertebral bodies (Fig 10B). In addition, there were signs of hypervascularisation (in the form of surface pitting and/or radial, superficial abnormal blood vessel impressions) on both acetabular fossa (Fig 12), around the margins of both acetabula, and around the neck of both femora (especially on the anterior and lateral aspects) (Fig 13). Finally, three types of stress indicators were also noted: the porotic type of *cribra orbitalia* in the left orbit (Fig 14B), the porotic type of *cribra cranii* on the left parietal bone (at the occipital angle) and the occipital bone (upper region of the squamous part), and signs of periostitis on the diaphysis of the left ulna, and both femora (Fig 14A) and tibiae (in the form of longitudinal striation).

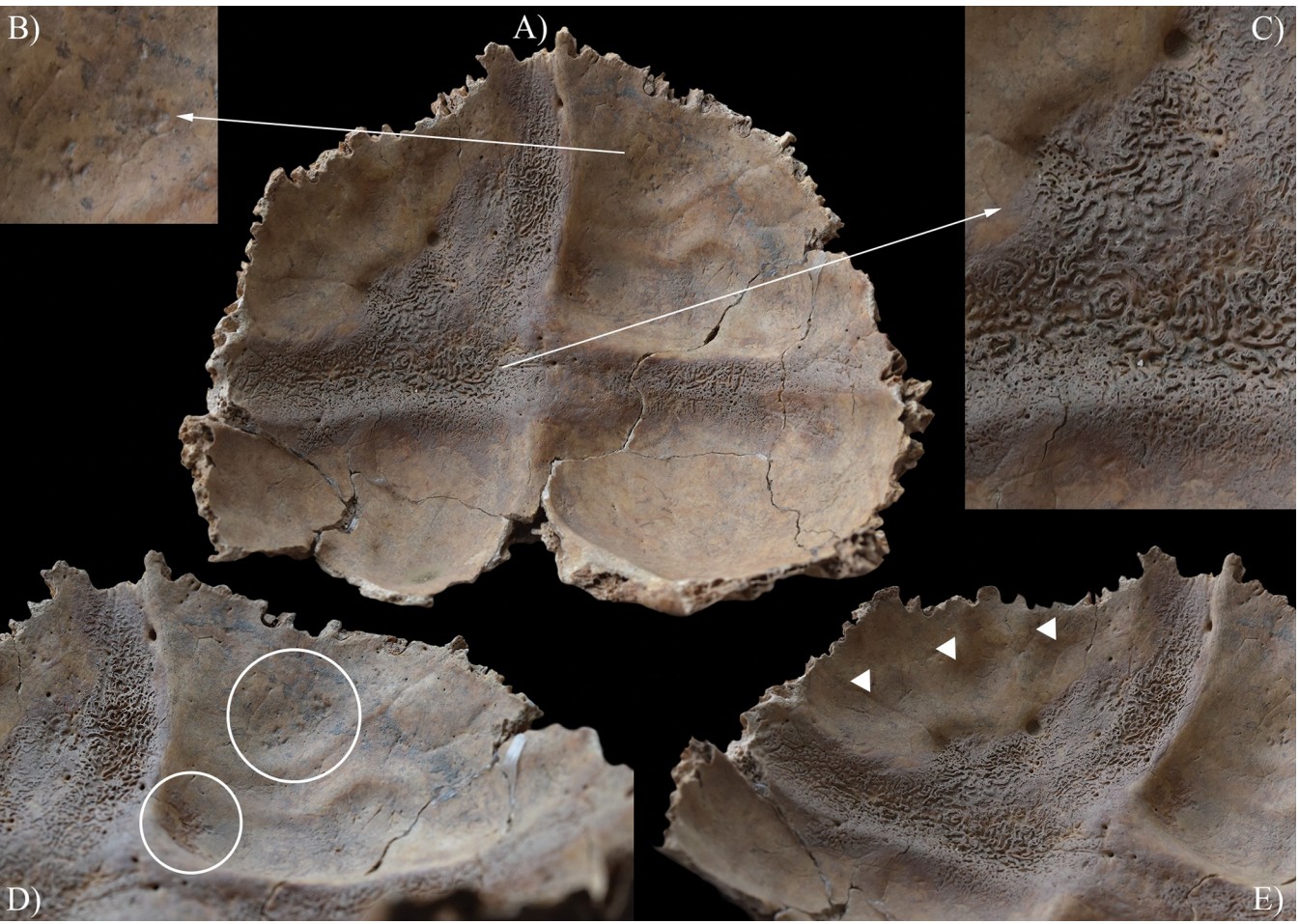

**Fig 6.** Endocranial granular impressions (left cerebral fossa), abnormal blood vessel impressions and periosteal appositions (sulci of the superior sagittal sinus and both transverse sinuses), and abnormally pronounced digital impressions (right cerebral fossa) on the occipital bone of HK225: A) squamous part, B) close-up of the granular impressions, C) close-up of the abnormal blood vessel impressions and periosteal appositions, D) granular impressions (white circles) in the left cerebral fossa, and E) abnormally pronounced digital impressions (white arrows) in the right cerebral fossa.

### HK253 (younger adult female)

During the macroscopic evaluation of the remains, only the inner surface of the cranium of **HK253** revealed alterations that are indicative of TB. Although the missing and taphonomic damage of the bones precluded the definitive observation of some regions of the endocranial surface (e.g., right parietal bone, and the squamous part of the occipital bone and right temporal bone), GIs were registered on the squamous part of the left temporal bone (Fig 15A and 15B), along the squamous suture on the left parietal bone (Fig 15D), and on the right greater wing of the sphenoid bone (Fig 15C). In addition, *cribra orbitalia* (porotic type) was detected in both orbits (Fig 16).

### HK309 (middle-aged to older adult male)

In the skeleton of **HK309**, only the endocranial surface revealed bony changes that can be ascribed to TB. The left parietal bone (along the squamous suture) (Fig 17A and 17B) and the right greater wing of the sphenoid bone (Fig 17C and 17D) presented GIs. Moreover, two types of stress indicators were noted: the porotic type of *cribra orbitalia* in both orbits

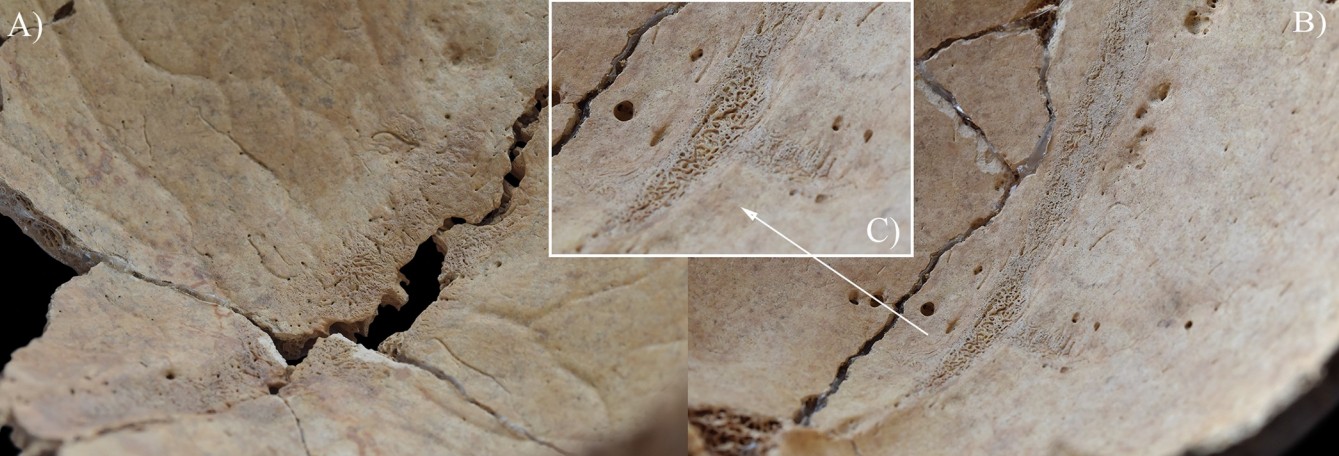

**Fig 7.** Endocranial abnormal blood vessel impressions and periosteal appositions along the sulcus of the superior sagittal sinus of HK225: A) left and right parietal bones, B) frontal bone, and C) close-up of the frontal bone.

(Fig 18B) and signs of periostitis on the lateral surface of the diaphysis of the right tibia (in the form of longitudinally striated subperiosteal new bone formations) (Fig 18A).

## Discussion and conclusions

### Endocranial bony changes

In **HK199**, **HK201**, **HK225**, **HK253**, and **HK309**, the presence of endocranial alterations (especially GIs) suggests that they suffered from TB meningitis (*leptomeningitis tuberculosa*) at the time of death. In all five cases, the pathogenic progression of this TB disease would have been that: via haematogenous or lymphogenous spread from the primary site of the infection (very likely the lungs), TB bacilli would have reached the central nervous system [66, 70–80]. Once there, the presence of the pathogens would have triggered the formation of small (0.5–2 mm) tubercles (i.e., Rich foci) in the meninges, and the brain and/or spinal cord parenchyma [66, 70–80]. Subsequently, TB bacilli would have been released into the cerebrospinal fluid (by rupture of one or more caseating Rich foci into the subarachnoid space or the ventricular system), triggering granulomatous inflammation of the leptomeninges (i.e., the *pia* and *arachnoid*

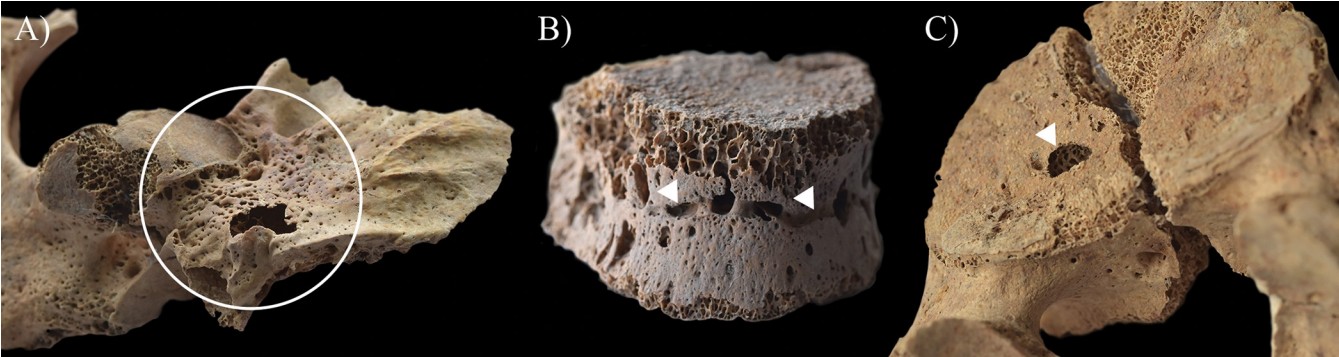

**Fig 8.** A) Pitting and slight cortical remodelling (white circle) on the right lateral part of the occipital bone of HK225; B) Signs of hypervascularisation (white arrows) on the anterior and lateral aspects of the T7 vertebral body of HK225; and C) An oval-shaped, well-circumscribed osteolytic lesion (white arrow) at the upper subchondral region of the L5 vertebral body of HK225.

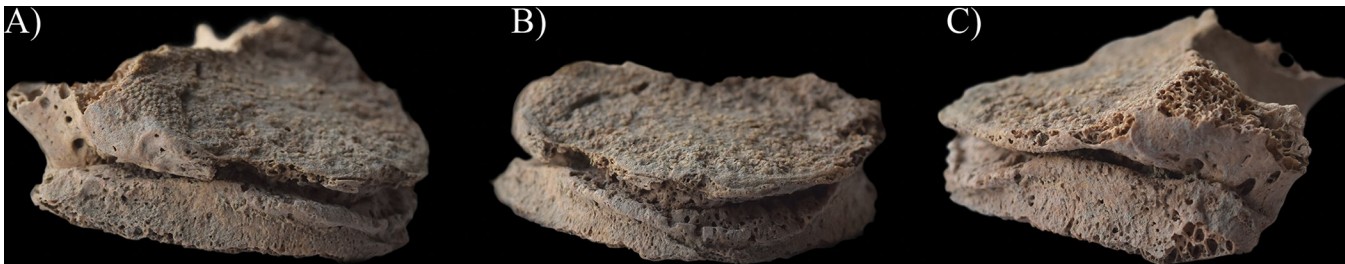

**Fig 9.** Almost complete destruction of the T2 vertebral body of HK225 with its consequent collapse due to which it became slightly wedge-shaped: A) right lateral view, B) anterior view, and C) left lateral view.

*mater encephali*) and the tubercle formation on them [66, 72–82]. It should be noted that besides the enhancing basal meningeal exudate and vasculitis of the blood vessels adjacent to or traversing the exudate, the meningeal tubercles are one of the characteristic pathological features of TB meningitis [66, 72, 75, 78, 81–83]. In later stages of the disease, not only the leptomeninges but also the outermost meningeal layer (i.e., the *dura mater encephali*) could have become affected by the pathological process [66, 84, 85]. As the *dura mater encephali* is directly adherent to the inner surface of the cranium, the gradually growing and coalescing meningeal tubercles could have exerted localised pressure on the underlying bone, inducing temporary, circumscribed bone atrophy and subsequent formation of endocranial GIs (Figs 2, 4, 5, 6A, 6B, 6D–6E, 15 and 17) [62, 65, 66, 69]. By definition, GIs are small (0.5–1.0 mm in diameter), relatively shallow (less than 0.5 mm in depth), roundish impressions with smooth margins and walls [62–66, 69]. They can appear separately or confluently and are usually grouped in clusters on the inner surface of the cranium [62–66, 69]. Granular impressions match in their position to those tubercles of the *dura mater encephali*, which cause bone resorption by exploring pressure on the underlying bone [62–66, 69]. It has been recently confirmed that GIs can be considered as specific signs of TB meningitis; and therefore, are sufficient enough on their own to make a definitive diagnosis of the disease in **HK199**, **HK201**, **HK225**, **HK253**, and **HK309** [62, 65, 66, 69].

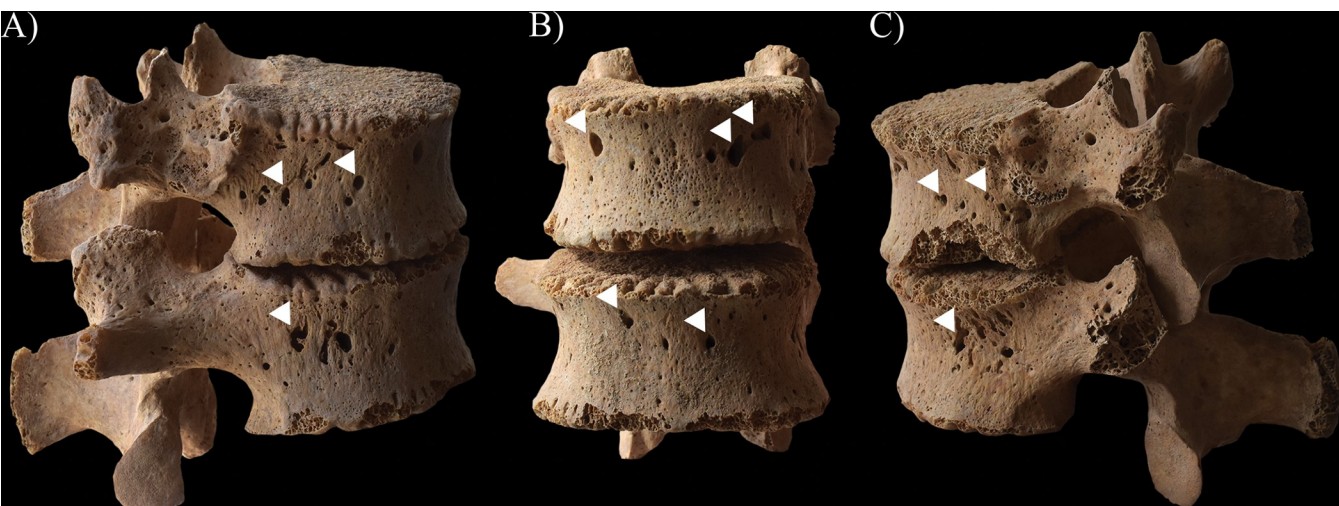

**Fig 10.** Signs of hypervascularisation (white arrows) on the anterior and lateral aspects of the T12–L1 vertebral bodies of HK225: A) right lateral view, B) anterior view, and C) left lateral view.

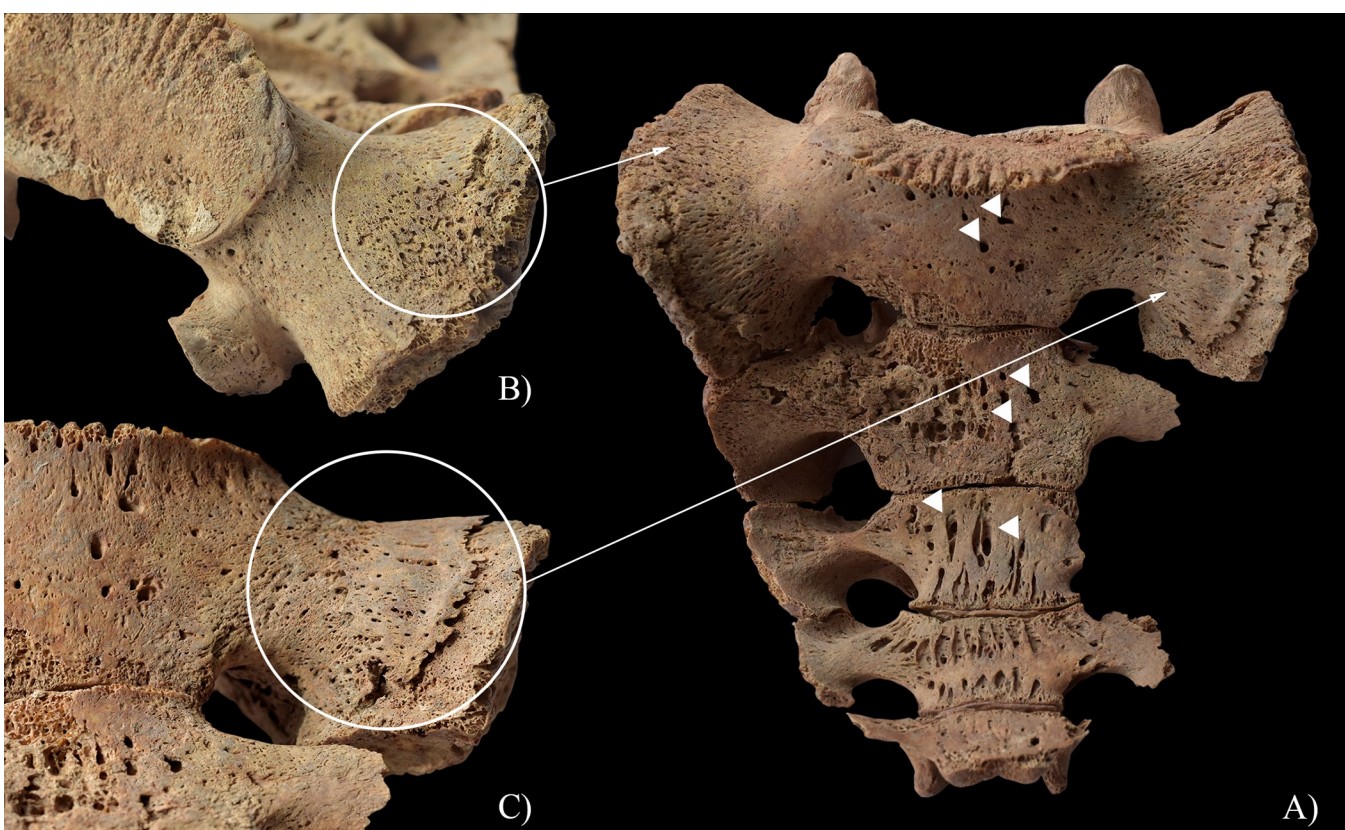

**Fig 11.** Signs of hypervascularisation (white arrows) A) on the pelvic surface of the sacrum of HK225, and surface pitting and reactive new bone formations (white circles) on the B) right and C) left sacral wings of HK225.

In two cases (**HK199** and **HK225**), besides GIs, other endocranial alterations indicative of TB meningitis (ABVIs, PAs, and/or APDIs) were observed. In both **HK199** and **HK225**, the presence of ABVIs and PAs (Figs 3, 6A, 5C–6E and 7) implies that not only the meninges but also at least some of the meningeal blood vessels and dural venous sinuses had become inflamed (i.e., vasculitis) [86–89]. This led to their disruption and secondary development of epidural haemorrhages [86–89]. As the epidural haemorrhages would have started to heal, intense vascularisation would have taken place (i.e., inflammatory phase) with subsequent formation of small, patch-like areas of very short, sinuous, branching ABVIs on the inner surface of the cranium [61, 63, 64, 67, 69]. In reparative phase of the healing process, there would have been deposition of newly formed, immature bone (i.e., woven bone) [61, 63, 64, 67, 69]. Thus, the endocranial ABVIs could have become more or less covered by PAs having a fibrous, porous, irregular, scab-like appearance [61, 63, 64, 67, 69]. In the last, remodelling phase of the healing process, an extensive, net-like aggregation of ABVIs would develop and the tongue-like PAs would have a very smooth, more mature appearance (i.e., lamellar bone) [61, 63, 64, 67, 69]. It should be noted that PAs could have occurred not only after the formation of epidural haemorrhages but also the granulomatous inflammation of the periosteal layer of the *dura mater encephali*, as this structure possesses an osteogenic potential [61, 63, 64, 67, 69, 90].

In **HK225**, a thick, gelatinous, basal inflammatory exudate would have formed in the initial stage of TB meningitis between the two layers of the leptomeninges [72–74, 77–78, 84, 91–104]. This exudate could have partially or completely filled the subarachnoid space and/or the ventricular pathways, with consequent development of progressive internal hydrocephalus

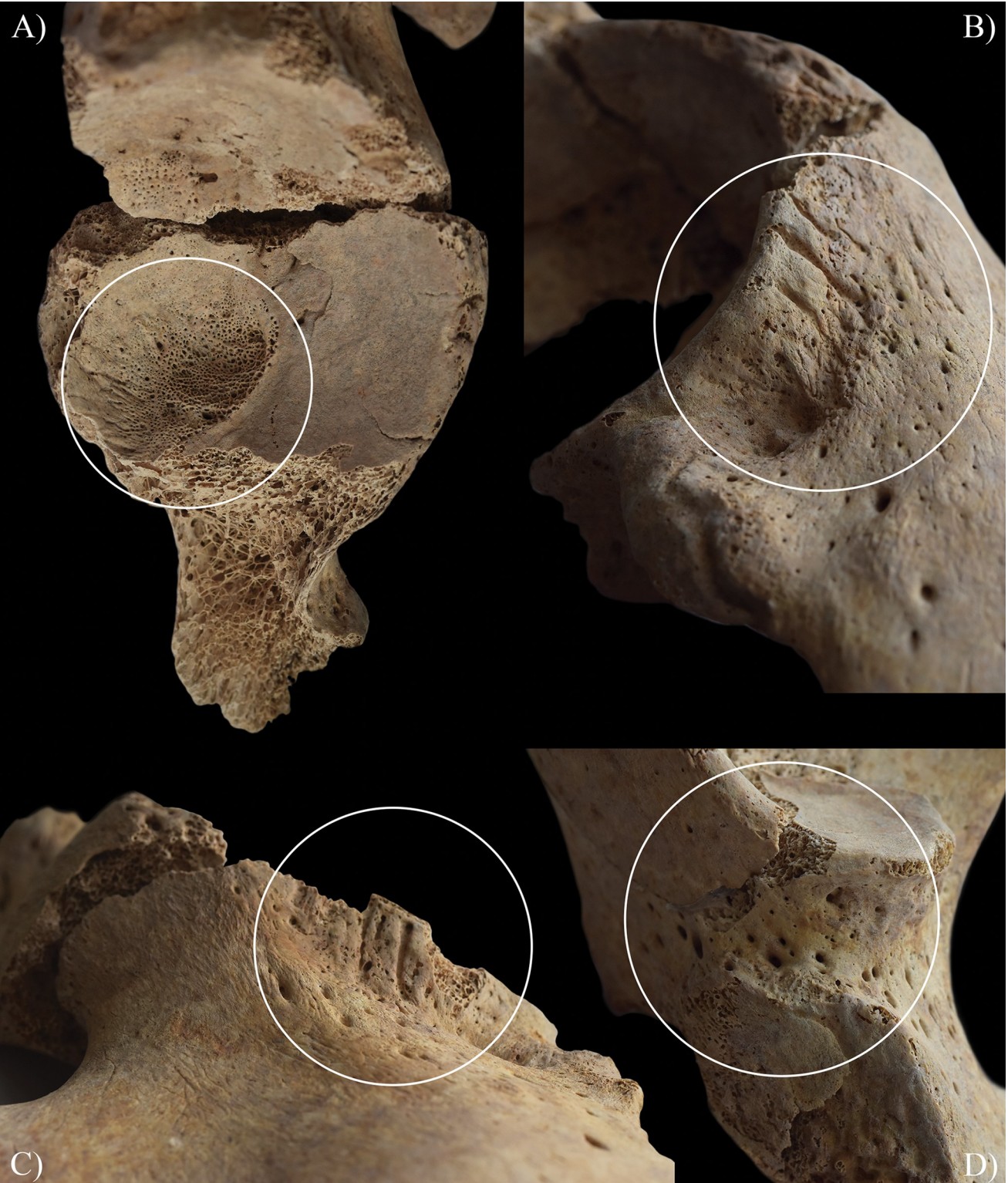

**Fig 12.** Signs of hypervascularisation (white circles) at both acetabula of HK225: A) pitting in the left acetabular fossa; pitting and radial, superficial abnormal blood vessel impressions on the superior margin of the B) right and C) left acetabula; and D) pitting at the inferior margin of the right acetabulum.

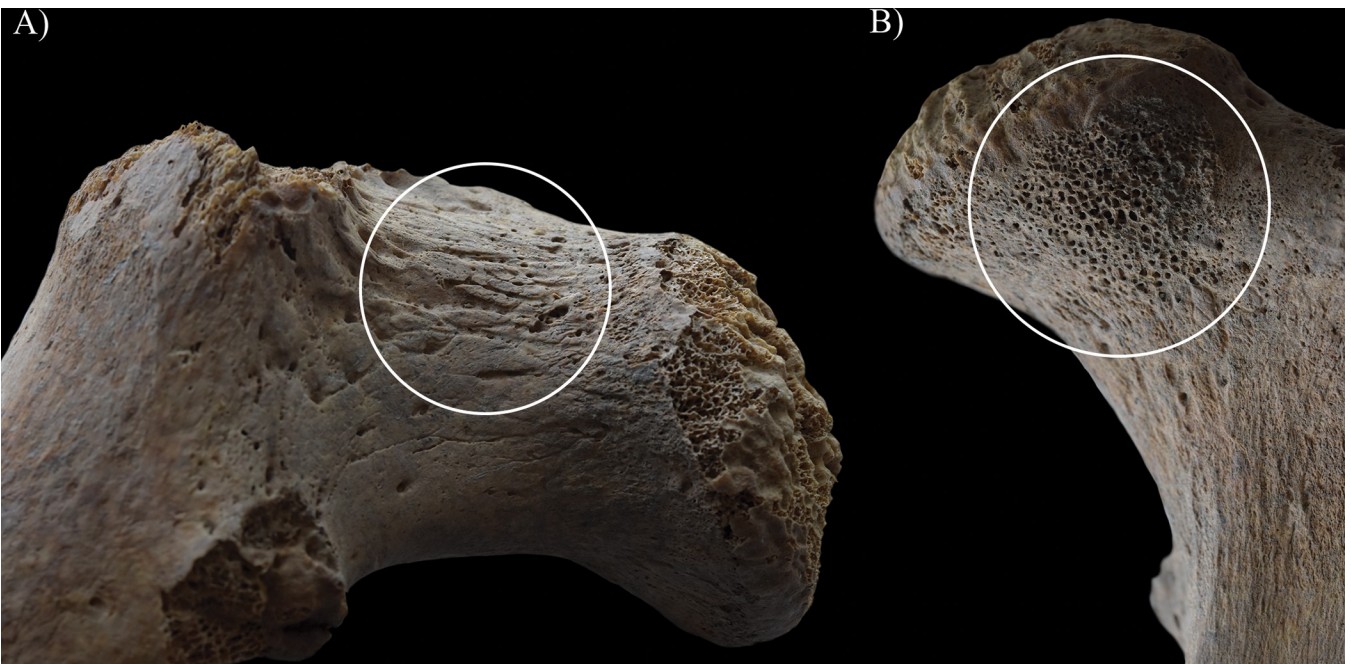

**Fig 13.** Signs of hypervascularisation (white circles) on the neck of the left femur of HK225: A) superficial abnormal blood vessel impressions on the lateral aspect and B) pitting on the anterior aspect.

[72–74, 77, 78, 84, 91–104]. Hydrocephalus is a disturbance in the production, flow or reabsorption of the cerebrospinal fluid, leading to an increase in the volume occupied by the cerebrospinal fluid in the central nervous system [102, 103]. Like many other cases with TB meningitis, in **HK225**, hydrocephalus would have been associated with a prolonged rise in the intracranial pressure [102, 103]. Due to the elevated intracranial pressure, the brain could have exerted localised pressure on the underlying bone, inducing temporary, circumscribed bone atrophy and subsequent formation of endocranial APDIs (Fig 6E) [61, 63, 68–69]. By definition, APDIs are shallow depressions resembling finger imprints [61, 63, 68, 69, 105–110]. They are incompletely separated from each other by bony ridges corresponding to cerebral sulci [61, 63, 68, 69, 105–110]. Abnormally pronounced digital impressions match in their position to those cerebral gyri, which cause bone resorption by exploring pressure on the underlying bone [61, 63, 68, 69, 105–110].

It is important to note that in contrast to GIs, the other three endocranial alteration types (ABVIs, PAs, and APDIs) observed in **HK199** and/or **HK225** are not pathognomonic for TB meningitis–they can be generated by many other infectious and non-infectious conditions. Besides TB, numerous other aetiologies should also be considered in the differential diagnosis of ABVIs, PAs, and APDIs. They can include: non-specific and specific meningitis, traumas, haemorrhages, brain tumours, and/or scurvy [61, 63, 64, 71, 83, 93, 95, 111–119]. Nevertheless, the concurrent presence of these endocranial alterations along with GIs in **HK199** and **HK225** indicates that the detected ABVIs, PAs, and APDIs are most likely resultant from TB meningitis [69].

## Non-endocranial bony changes

From the five examined individuals, only **HK225** presented non-endocranial bony changes that are indicative of TB. The skeletal lesions observed in the spine and both hip joints of this

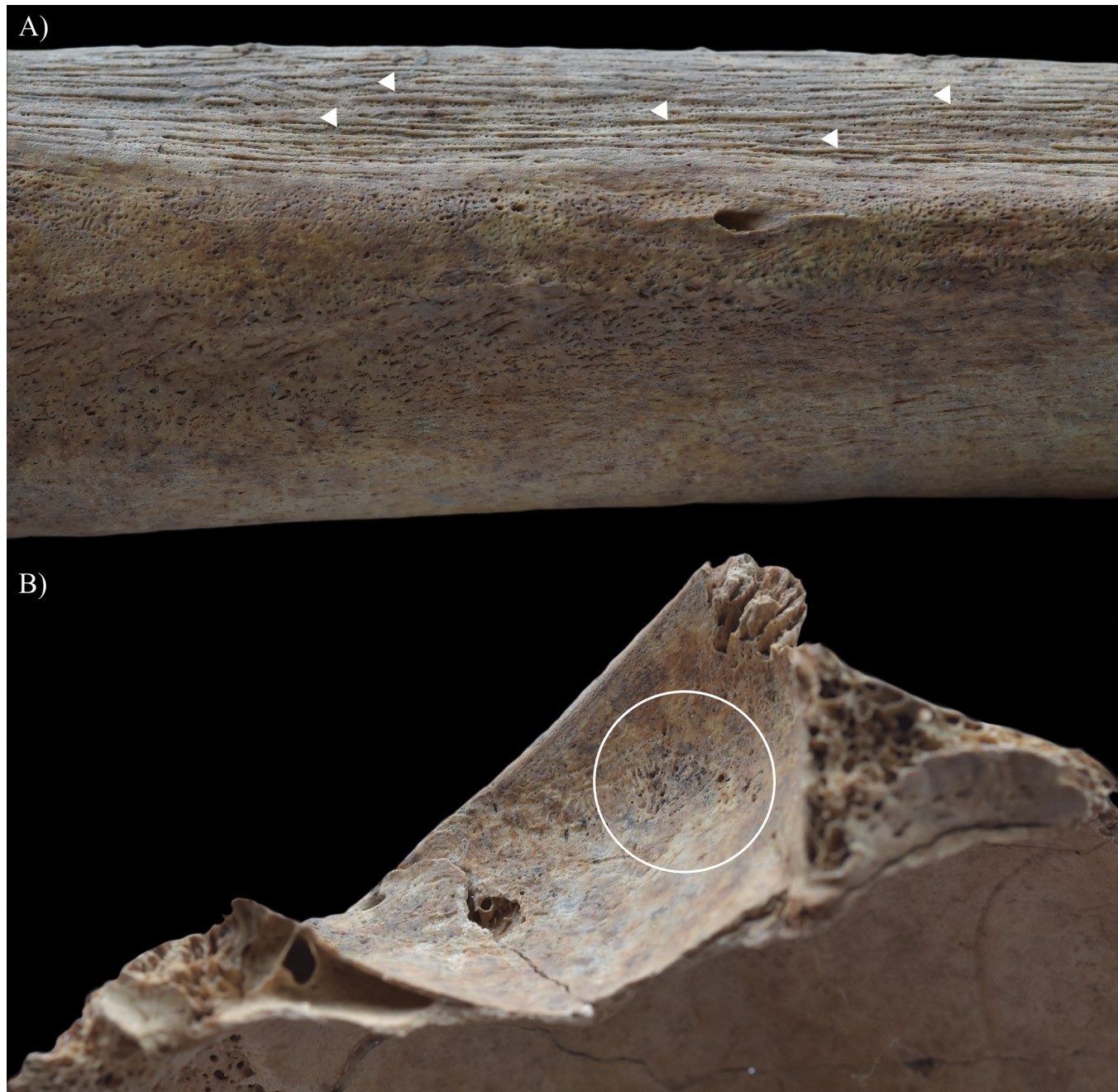

**Fig 14.** A) Signs of periostitis (white arrows) on the diaphysis of the left femur (medial surface) of HK225 and B) Porotic type of *cribra orbitalia* (white circle) in the left orbit of HK225.

juvenile individual can be associated with multifocal osteoarticular TB. The step-by-step pathogenesis of this TB disease would have been that: via haematogenous or lymphogenous spread from the primary site of the infection (very likely the lungs), TB bacilli would have been deposited in the spine and both hip joints, triggering granulomatous inflammation [120–128]. Based on the severity and extent of the alterations detected in the vertebral column of **HK225**, the earliest site of the TB infection may have been the T2 vertebral body. On the one hand,

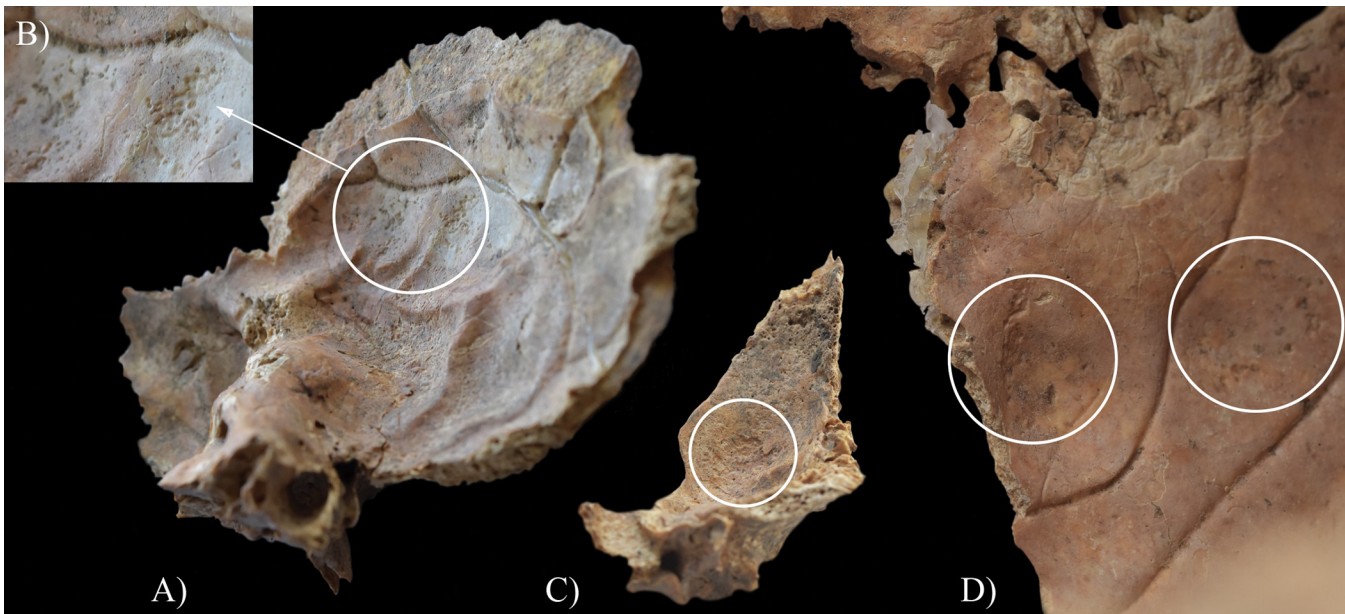

**Fig 15.** Granular impressions (white circles) on the endocranial surface of HK253: A) left temporal bone (squamous part), B) close-up of the left temporal bone, C) right greater wing of the sphenoid bone, and D) left parietal bone (close to the squamous suture).

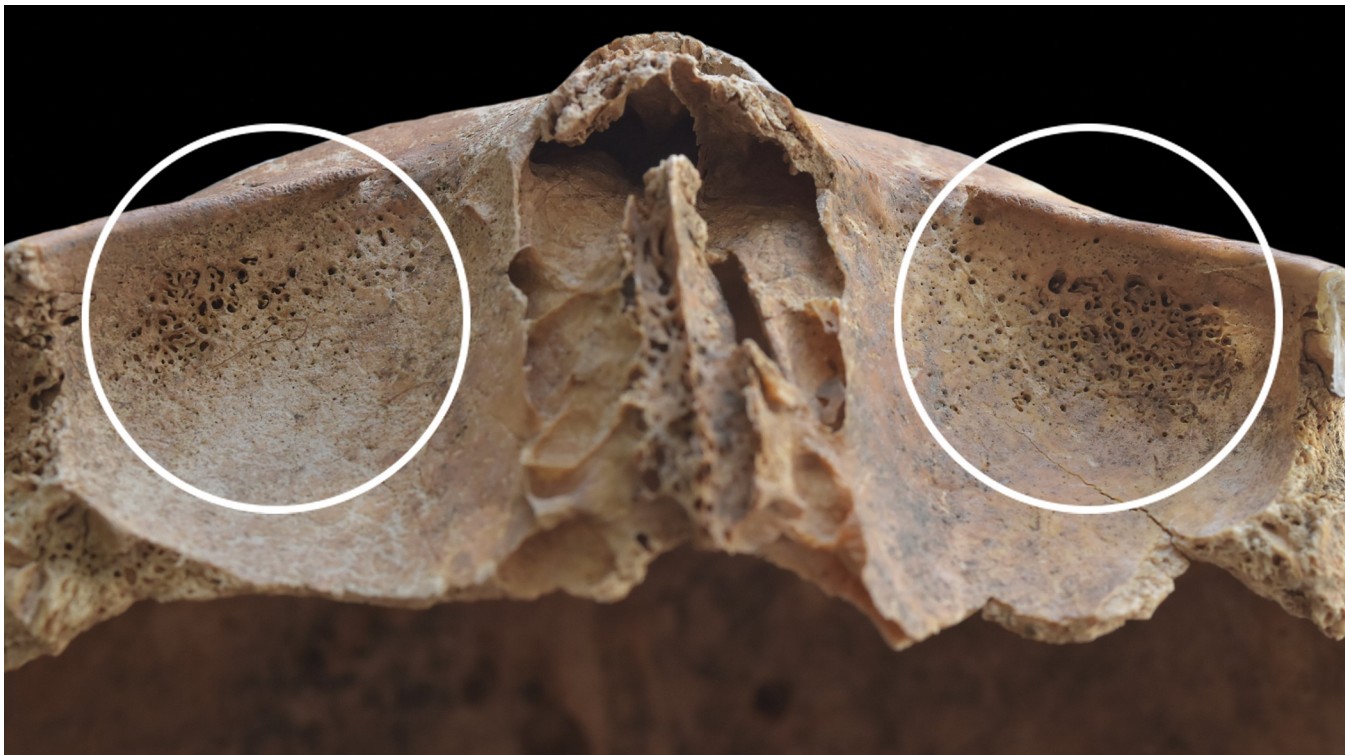

**Fig 16. Porotic type of *cribra orbitalia* (white circles) in both orbits of HK253.**

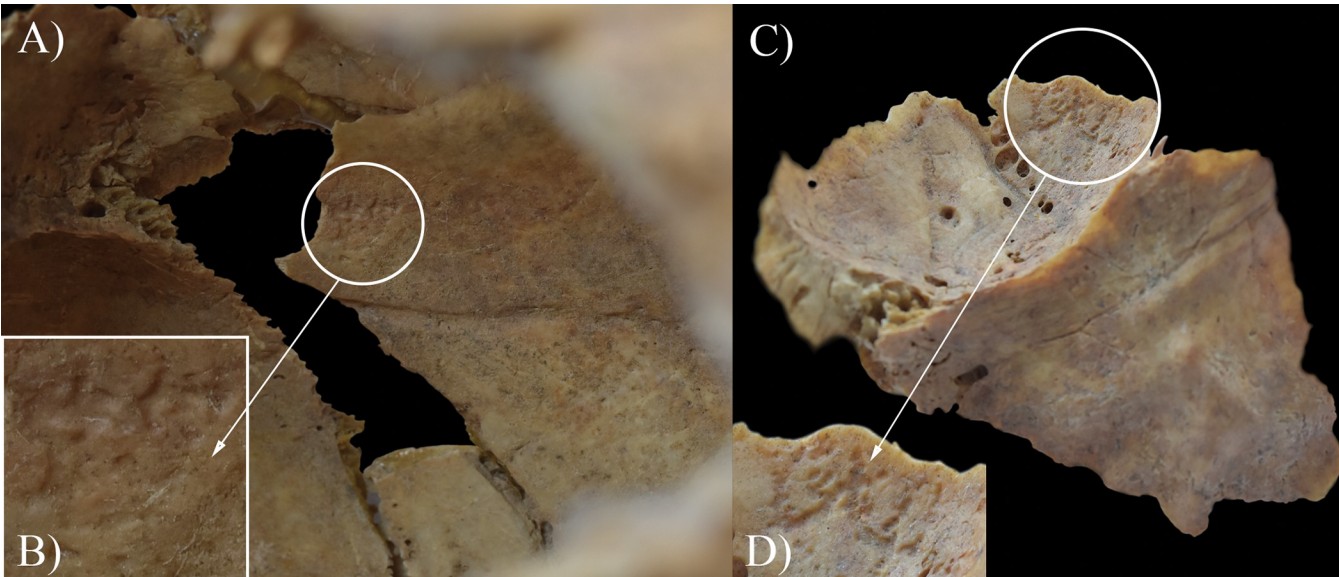

**Fig 17.** Granular impressions (white circles) on the endocranial surface of HK309: A) left parietal bone (close to the squamous suture), B) close-up of the left parietal bone, C) right greater wing of the sphenoid bone, and D) close-up of the right greater wing of the sphenoid bone.

gradually expanding and coalescing tubercles could have developed in the bone marrow of T2; subsequently, their caseous necrosis would have ensued [1, 120, 126, 129–134]. On the other hand, there could have been TB involvement of the respective segmental artery branches with secondary ischaemia [1, 120, 126, 129–134]. These two pathological processes would have resulted in caseous and/or ischaemic bone necrosis, and secondary resorption of the bone trabeculae [1, 120, 126, 129–134]. Later, formation of more and more osteolytic lesions in the cancellous bone gave rise to progressive destruction and consequent gradual weakening of the T2 vertebral body [1, 12, 120, 126, 133, 135–140]. Finally, it collapsed under stress placed upon it by gravity and muscle activity, leading to its slightly wedge-shaped appearance (Fig 9) [1, 12, 120, 126, 133, 135–140]. In L5, the presence of an oval-shaped, well-circumscribed osteolytic lesion at the upper subchondral region (Fig 8C) indicates that its vertebral body became an additional site of the TB infection in the spine. However, **HK225** likely died before more severe bone destruction could have taken place. Based on the signs of hypervascularisation on the anterior and/or lateral aspects of the lower cervical (C5–7), thoracic (T1 and T3–12), and lumbar (L1–5) vertebral bodies, as well as the pelvic surface of the sacrum (S1–4) (Figs 8B, 10 and 11A), it cannot be excluded that these spinal regions were also involved by TB. The enlarged vascular foramina have been described as probable signs of early-stage spinal TB in other palaeopathological studies [49, 50, 141]. It is important to note that these alterations are not pathognomonic for spinal TB, because other medical conditions (e.g., ankylosing spondylitis or brucellosis), or in subadults like **HK225**, even the normal spinal development can result in vertebral hypervascularisation [49, 50]. In **HK225**, the presence of surface pitting and/or slight cortical remodelling on the anterior aspect of some lower thoracic and lumbar vertebral bodies, as well as the ectocranial surface of the occipital bone (on both lateral parts around the occipital condyles) (Figs 8A and 10B) supports that the TB infection was spreading vertically along the spine (both upwards and downwards), probably beneath the anterior longitudinal ligament. Likely, the presence and extension of an extra-vertebral TB abscess in the sub-ligamentous space resulted in the development of the skeletal lesions (by combined pressure and ischaemic effects) [1, 12, 120, 123, 129, 130, 132, 140]. There may have also been the spread of

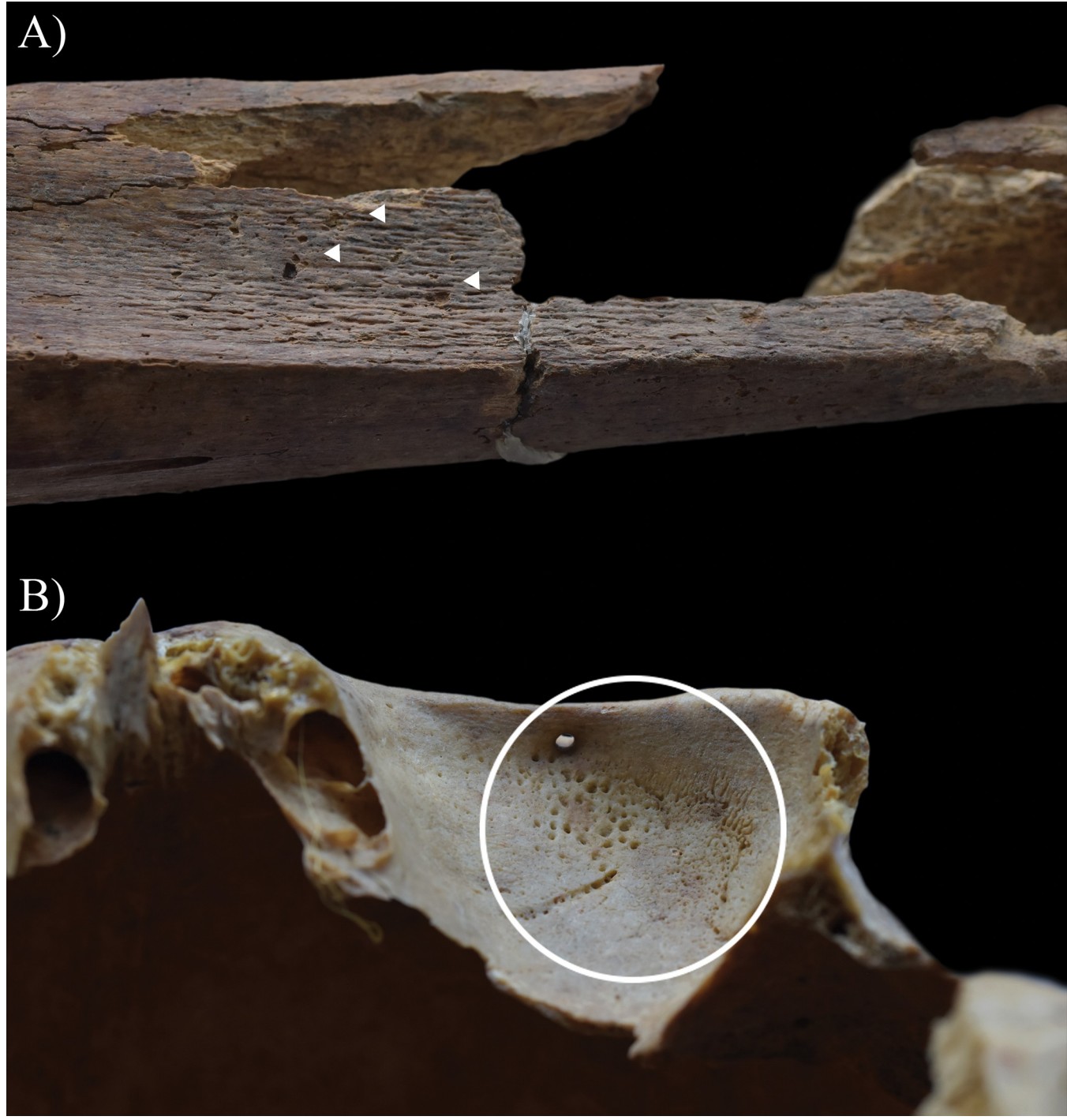

**Fig 18.** A) Signs of periostitis (white arrows) on the diaphysis of the right tibia (lateral surface) of HK309 and B) Porotic type of *cribra orbitalia* (white circle) in the left orbit of HK309.

the TB infection from the diseased vertebrae of **HK225** into the adjacent soft tissues (e.g., muscles and ligaments), leading to formation of an extra-vertebral cold abscess (a slow-growing abscess without characteristic signs of inflammation, such as heat, erythema, or tenderness) [1, 12, 122, 138]. The TB mass could have accumulated in the pelvic area–the surface pitting and

reactive new bone formations recorded on the sacral wings of **HK225** (Fig 11) likely developed in response to the presence of this overlying cold abscess [1, 12, 122, 138]. Numerous differential diagnoses, such as pyogenic spinal infections and granulomatous spinal infections other than TB (e.g., brucellosis or aspergillosis) (S2 Text) [122, 142–145], need to be considered for the bony changes observed in the vertebral column of **HK225**. Based on their nature, association, and distribution pattern, they are most consistent with spinal TB. Their concurrent presence along with endocranial alterations indicative of TB meningitis further supports their tuberculous origin in **HK225**.

Not only the vertebral column of **HK225** but also the hip joints revealed signs of hypervascularisation (i.e., surface pitting and/or radial superficial abnormal blood vessel impressions) around the margins of both acetabula, on both acetabular fossae, and around the neck of both femora (especially on the anterior and lateral aspects) (Figs 12 and 13). This localised peri-articular osteoporosis is a characteristic feature of the initial or 'synovitis' stage of TB arthritis [127, 146–151]. Lodgement of TB bacilli into the synovium of the hip joints of **HK225** would have triggered granulomatous inflammation (i.e., synovitis) [127, 146–151]. Likely, this gave rise to increased synovial vascularisation (i.e., hyperaemia), leading to the development of the bony changes in both innominate bones and femora [127, 146–151]. Based on the concurrent presence of localised peri-articular osteoporosis along with endocranial alterations indicative of TB meningitis and with vertebral lesions indicative of spinal TB, TB arthritis seems to be its most likely underlying cause in **HK225**. Other aetiologies could still be considered in the differential diagnosis, such as rheumatoid arthritis and pyogenic arthritis (S3 Text) [152–156].

## Presence of tuberculosis in the Sarmatian-period Barbaricum of the Carpathian Basin

Until now, only three TB cases have been reported from the Sarmatian-period Barbaricum of the Carpathian Basin [22]. All of them derive from the 4th–5th-century-CE archaeological site of Apátfalva–Nagyút dűlő, located in the Tisza-Maros region, about 40 km far from the Hódmezővásárhely–Kenyere-ér, Bereczki-tanya archaeological site. To compare the five newly discovered TB cases with the three from the Apátfalva–Nagyút dűlő archaeological site (grave no. 178 –**AN178**, grave no. 189 –**AN189**, and grave no. 195 –**AN195**), these three skeletons were re-evaluated using the same palaeopathological diagnostic criteria for TB that were applied to the examination of the Hódmezővásárhely–Kenyere-ér, Bereczki-tanya osteoarchaeological series.

In the two sites, the demographics of TB cases are divergent from one another. Of the three TB cases from Apátfalva–Nagyút dűlő, one was an adult individual of indetermined sex (**AN189**; 20≤ years old) and two were children–**AN178** (c. 8 years old) and **AN195** (c. 7 years old). Of the five TB cases from Hódmezővásárhely–Kenyere-ér, Bereczki-tanya, four were adults (two females, one male, and one of indetermined sex) and one was a c. 14–16 years old individual.

The manifestation of TB is different between the series. In Hódmezővásárhely–Kenyere-ér, Bereczki-tanya, all five cases exhibit endocranial alterations, indicating that all of them suffered from TB meningitis at the time of death. In Apátfalva–Nagyút dűlő, none of the examined individuals display TB-related bony changes on the inner surface of the cranium. All three TB cases from Apátfalva–Nagyút dűlő only represent the osteoarticular form of TB–**AN178** and **AN195** had spinal TB, and **AN189** had TB arthritis of both sacroiliac joints. In contrast, only **HK225** show skeletal lesions suggestive of multifocal osteoarticular TB (spine and both hip joints). **HK225** had almost the whole vertebral column involved by TB infection, but **AN178** and **AN195** only had the cervico-thoracic and lumbar regions affected. In **AN178**, the

pathological process involved not only one but six contiguous vertebrae (C4–T2), with destruction and collapse of the vertebral bodies. Not only the body remnants but also the posterior elements of the diseased vertebrae fused together, leading to a Pott's gibbus. Besides the severe angular kyphosis in the cervico-thoracic spine, there was also dextroscoliosis. The remnants of the C6–7 vertebral bodies were dislocated to the right side of the T2 vertebral body. Based on the macroscopic characteristics of the observed vertebral changes, **AN178** can represent the paradiscal form of spinal TB, similar to that found in **HK225**. In **AN195**, there were no osteolytic lesions but only some erosive changes on the anterior aspect of the vertebral bodies in the lumbar region of the spine. The pathological process affected not only one but at least three vertebrae of which the exact location in the lumbar spine cannot be determined. Based on the macroscopic characteristics of the recorded vertebral alterations, **AN195** can represent not the paradiscal but the anterior sub-ligamentous form of spinal TB. In **AN189**, TB did not affect the vertebral column but both sacroiliac joints. Slight erosive changes were detected in the central area of the auricular surfaces, and the right innominate bone has extensive erosion of the auricular surface with its almost complete destruction. The left innominate bone was absent from the skeleton. It is important to note that the missing and taphonomic damage of the bone remains of **AN178**, **AN189**, and **AN195** significantly hindered the macromorphological observations, so other parts of the skeleton may have also been affected by TB. On the other hand, the Apátfalva–Nagyút dűlő osteoarchaeological series has been evaluated only for osteoarticular TB by Marcsik & Kujáni [22], so other forms of TB may have also been present in this Sarmatian-period population. Based on the above, systematic re-evaluation of the entire series would be desirable.

This research highlights the importance of the development in the palaeopathological diagnostic techniques of TB. Initially, the retrospective diagnosis of the disease was based on the identification of macroscopic pathological bony changes in the skeleton that have been found to be associated with different forms of osteoarticular TB (e.g., spinal TB or TB arthritis of the large, weight-bearing joints) [1, 12, 50, 58, 66]. The problem is that osteoarticular TB (an extra-pulmonary form of the disease that directly affects the skeleton) is very rare–only comprising about 1–3% of patients with active TB disease today [66, 120, 124, 130, 157]. Consequently, if we only consider those alterations that can be related to the different forms of osteoarticular TB, we will invariably overlook numerous cases [66]–such as seen in the Apátfalva–Nagyút dűlő osteoarchaeological series. Since the 1980s, several studies [7, 50–54, 57–58, 60–62, 65–69] have been carried out to identify further macroscopic diagnostic criteria for TB. As a result of these projects, a positive association has been recognised between some pathological alterations and different extra-skeletal forms of TB, which can indirectly affect the bones with development of skeletal lesions. For instance, periosteal new bone formations on the visceral surface of ribs and signs of diffuse, symmetrical periostitis on the diaphysis of the short and long tubular bones are related to pulmonary TB/TB pleurisy [51–54, 57, 58], and endocranial GIs, ABVIs, PAs, and APDIs are associated with TB meningitis [60–62, 65–69]. Despite the missing and taphonomic damage of the bone remains in **HK199**, **HK201**, **HK225**, **HK253**, and **HK309**, the preservation of their crania was sufficient to evaluate them for the presence of GIs, ABVIs, PAs, and APDIs, allowing the determination of TB infection in these five cases.

In summary, **HK199**, **HK201**, **HK225**, **HK253**, and **HK309** provide us invaluable information about the spatio-temporal distribution of TB in the past. Thanks to their discovery, the number of TB cases known from the Sarmatian-period Carpathian Basin has doubled, suggesting that the disease was likely more frequent in the Barbaricum (at least in the Tisza-Maros region) than previously thought. Without the application of GIs, the diagnosis of TB could not have been established in these five cases. Thus, the identification of TB in **HK199**, **HK201**,

**HK225**, **HK253**, and **HK309** highlights the importance of diagnostics development, especially the refinement of diagnostic criteria. From the above findings, the systematic macromorphological (re-)evaluation of osteoarchaeological series from the Sarmatian-period Carpathian Basin would be desirable, as we could get a more accurate picture of the burden that TB may have put on the ancestral human communities of the Barbaricum.

## Supporting information

**S1 Text. The Sarmatian period ($1^{st}$–$5^{th}$ centuries CE) in the Barbaricum of the Carpathian Basin.**
(PDF)

**S2 Text. Differential diagnoses of the skeletal lesions indicative of tuberculous involvement of the spine that were detected in HK225.**
(PDF)

**S3 Text. Differential diagnoses of the bony changes indicative of tuberculous involvement of both hip joints that were observed in HK225.**
(PDF)

## Author Contributions

**Conceptualization:** Olga Spekker.

**Data curation:** Olga Spekker, Sándor Varga, Balázs Tihanyi.

**Formal analysis:** Olga Spekker, Balázs Tihanyi.

**Funding acquisition:** Olga Spekker, Luca Kis, Tibor Török.

**Investigation:** Olga Spekker, Kitty Király, Sándor Varga, Antónia Marcsik, Oszkár Schütz, Balázs Tihanyi.

**Methodology:** Olga Spekker, Balázs Tihanyi.

**Project administration:** Olga Spekker.

**Resources:** Tibor Török.

**Supervision:** David R. Hunt.

**Visualization:** Olga Spekker, Luca Kis.

**Writing – original draft:** Olga Spekker, Attila Kiss P., Balázs Tihanyi.

**Writing – review & editing:** Olga Spekker, David R. Hunt.

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
