## [Decision Letter · Decision Letter 0]

24 Jul 2023

PONE-D-23-20370White Plague among the “Forgotten People” from the Barbaricum of the Carpathian Basin – Cases with tuberculosis from the Sarmatian-period (3rd–4th centuries CE) archaeological site of Hódmezővásárhely–Kenyere-ér, Bereczki-tanya (Hungary)PLOS ONE

Dear Dr. Spekker,

Thank you for submitting your manuscript to PLOS ONE. After careful consideration, we feel that it has merit but does not fully meet PLOS ONE’s publication criteria as it currently stands. Therefore, we invite you to submit a revised version of the manuscript that addresses the points raised during the review process.

We look forward to receiving your revised manuscript.

Kind regards,

Dario Piombino-Mascali, Ph.D.

Academic Editor

PLOS ONE

https://www.sciencedirect.com/science/article/abs/pii/S147297922200124X?via%3Dihub

https://journals.plos.org/plosone/article?id=10.1371%2Fjournal.pone.0230418

In your revision ensure you cite all your sources (including your own works), and quote or rephrase any duplicated text outside the methods section. Further consideration is dependent on these concerns being addressed.

4. In your manuscript, please provide additional information regarding the specimens used in your study. Ensure that you have reported human remain specimen numbers and complete repository information, including museum name and geographic location.

For more information on PLOS ONE's requirements for paleontology and archeology research, see https://journals.plos.org/plosone/s/submission-guidelines#loc-paleontology-and-archaeology-research.

5. We note that Figure 1 in your submission contain copyrighted images. All PLOS content is published under the Creative Commons Attribution License (CC BY 4.0), which means that the manuscript, images, and Supporting Information files will be freely available online, and any third party is permitted to access, download, copy, distribute, and use these materials in any way, even commercially, with proper attribution. For more information, see our copyright guidelines: http://journals.plos.org/plosone/s/licenses-and-copyright.

6. We note that Figures 2, 3, 4, 5, 6, 7, 8, 9, 10, 11, 12, 13, 14, 15, 16 , 17 and 18 in your submission contain [map/satellite] images which may be copyrighted. All PLOS content is published under the Creative Commons Attribution License (CC BY 4.0), which means that the manuscript, images, and Supporting Information files will be freely available online, and any third party is permitted to access, download, copy, distribute, and use these materials in any way, even commercially, with proper attribution. For these reasons, we cannot publish previously copyrighted maps or satellite images created using proprietary data, such as Google software (Google Maps, Street View, and Earth). For more information, see our copyright guidelines: http://journals.plos.org/plosone/s/licenses-and-copyright.

1. You may seek permission from the original copyright holder of Figures 2, 3, 4, 5, 6, 7, 8, 9, 10, 11, 12, 13, 14, 15, 16 , 17 and 18 to publish the content specifically under the CC BY 4.0 license. 

Additional Editor Comments:

Dear Dr Spekker,

I have received the comments from the reviewers, and I am happy to reconsider the paper after the minor revisions are carried out. I also kindly ask you to do the following:

1) Please double check that all references are correctly formatted according to the style of Plos one;

2) Please use just either British or American spelling but not a mixture of the two;

3) Please add to the ethics statement some consideration regarding the fact that the remains were treated with dignity and respect during investigation, and consider adding, as a reference, the ethical recommendation made by Squires et al. released in 2022 on the AJBA;

4) Please have the manuscript read by a native English speaker prior to re-submission, which is going to help for clarity.

I look forward to receiving your revised manuscript.

Reviewers' comments:

Reviewer's Responses to Questions

**Comments to the Author**

1. Is the manuscript technically sound, and do the data support the conclusions?

Reviewer #1: Partly

Reviewer #2: Yes

2. Has the statistical analysis been performed appropriately and rigorously? 

Reviewer #1: N/A

Reviewer #2: Yes

3. Have the authors made all data underlying the findings in their manuscript fully available?

Reviewer #1: Yes

Reviewer #2: Yes

4. Is the manuscript presented in an intelligible fashion and written in standard English?

Reviewer #1: Yes

Reviewer #2: Yes

5. Review Comments to the Author

Reviewer #1: Dear Editor of PLOS ONE, Professor Dario Piombino-Mascali, and Authors,

I have read the manuscript titled « White Plague among the “Forgotten People” from the Barbaricum of the Carpathian Basin – Cases with tuberculosis from the Sarmatian-period (3rd–4th centuries CE) archaeological site of Hódmezővásárhely–Kenyere-ér, Bereczki-tanya (Hungary)» and I have found it interesting and relevant, well-written and researched, providing new knowledge about the possible presence of tuberculosis (TB) in a past community of Hungary. There are some questions with the paper that I would like to see addressed by the authors.

1. Paleopathology does not always focus on the individual, as it can focus – and several times it does – in population samples; it seeks a differential diagnosis, not a «definite diagnosis» - which is impossible to attain most of the times.

2. Age at death and biological sex were estimated through different standard methods but, for each individual skeleton, we do not know which methods were used to estimate those parameters. That information must be added to each individual paleodemographic details (page 16 and after, line 232 and after). Please remove the tag «object» to the individuals, human skeletal remains are not objects and we shouldn’t reify that.

3. Except for the juvenile individual, a TB diagnosis was achieved based solely on intracranial lesions. Even though there are good signs that TB is associated with granular impressions in the inner skull surfaces, there are also competing hypotheses for these lesions that must be handled in a differential diagnosis, especially because no other TB-related lesions were observed.

My best regards.

Reviewer #2: Olga Spekker and colleagues subjected in their study “White Plague among the “Forgotten People” from the Barbaricum of the Carpathian Basin – Cases with tuberculosis from the Sarmatian-period (3rd–4th centuries CE) archaeological site of Hódmezővásárhely–Kenyere-ér, Bereczki-tanya (Hungary) 28 skeletal individuals from the Sarmatian period (1st–5th centuries CE) in the Barbaricum of the Carpathian Basin to a detailed paleopathological evaluation. Thereby they identified 5 individuals that display bone changes indicative for Tuberculosis meningitis. One of the five TB cases displayed in addition skeletal signs for osteoarticular TB. Overall, the methods applied in this study are well described and the obtained results are interesting for the fields of anthropology, paleopathology, archaeology, historical research, and paleomicrobiology.

Beside one minor comment I would like to ask the authors to comment in the manuscript in more detail on the possibility that the observed skeletal changes can derive from infectious/metabolic disease others than TB. I assumed so far that there exists only one clear cut clinical picture in skeletal remains for TB, the Potts disease. All other skeletal changes linked to TB (e.g. on ribs or the skull) are indicative for this disease but it cannot be excluded that these changes are caused by other diseases. I am not a paleopathologist so please correct me if I am wrong.

I saw that you also talk in your paleopathological methods part of skeletal alterations that are “indicative of TB” (line 220, 223) or “suggestive of TB meningitis” (line 226). If these are first indications of TB in these individuals than I would like to ask you to also consider this throughout your manuscript by:

- Toning down the title “Presence of tuberculosis in the Sarmatian-period…” in the discussion

- And by including a paragraph in the discussion highlighting the possibility that these changes could potentially also be caused by other diseases.

Minor comments:

I would like to ask the authors to shorten the introduction of the Sarmatian-period in the introductory part (line 85 to 150). This very detailed description distracts in my opinion a bit from the actual topic of the paper.

6. PLOS authors have the option to publish the peer review history of their article (what does this mean?). If published, this will include your full peer review and any attached files.

Reviewer #1: No

Reviewer #2: No

---

## [Author Response · Author response to Decision Letter 0]

5 Oct 2023

Dear Dr. Piombino-Mascali,

We are very thankful for your and the reviewers’ insightful and constructive comments regarding our manuscript entitled “White Plague among the “Forgotten People” from the Barbaricum of the Carpathian Basin – Cases with tuberculosis from the Sarmatian-period (3rd–4th centuries CE) archaeological site of Hódmezővásárhely–Kenyere-ér, Bereczki-tanya (Hungary)” that was submitted to PLOS ONE (manuscript ID: PONE-D-23-20370). We are sure that you and the reviewers helped us to improve the quality of our manuscript. The main text has been modified following the suggestions, and three supplementary texts have been created and added. The revised files have been uploaded to the submission site of the journal.

Responses to the reviewers’ comments:

1) Reviewer 1 commented that “I have read the manuscript titled « White Plague among the “Forgotten People” from the Barbaricum of the Carpathian Basin – Cases with tuberculosis from the Sarmatian-period (3rd–4th centuries CE) archaeological site of Hódmezővásárhely–Kenyere-ér, Bereczki-tanya (Hungary)» and I have found it interesting and relevant, well-written and researched, providing new knowledge about the possible presence of tuberculosis (TB) in a past community of Hungary. There are some questions with the paper that I would like to see addressed by the authors.”. 

We are very grateful for Reviewer 1 for this positive assessment of our manuscript. We appreciate the points they have raised and we have responded to each below.

2) Reviewer 1 mentioned that “Paleopathology does not always focus on the individual, as it can focus – and several times it does – in population samples; it seeks a differential diagnosis, not a «definite diagnosis» - which is impossible to attain most of the times.”.

We thank Reviewer 1 for nothing these important points. In Milner and Boldsen (2017) (https://www.sciencedirect.com/science/article/abs/pii/S1879981717300487?via%3Dihub – reference no. 4 in the originally submitted and revised versions), the authors have discussed about the differences between palaeopathology and palaeoepidemiology:

“Paleoepidemiology is a relatively new approach to characterizing the life experiences of prehistoric and historic-period peoples (Boldsen, 2001, Boldsen and Milner, 2012, Dutour, 2008, Pinhasi and Turner, 2008, Waldron, 1994, Waldron, 2007). While closely related to paleopathology because it is based on sound observations of disease processes in skeletons, paleoepidemiology employs different analytical procedures and has distinctive objectives. … At the outset, it is important to realize that populations, not individuals, are of interest. That distinguishes paleoepidemiology from case studies featuring one or a few skeletons, which are common in paleopathological research.” (page 26 – Introduction section).

In another article published by Dangvard Pedersen et al. (2019) (https://www.sciencedirect.com/science/article/abs/pii/S1879981718301608?via%3Dihub – reference nos. 6 and 7 in the originally submitted and revised versions, respectively), the authors have made a similar distinction between palaeopathology and palaeoepidemiology:

“Tuberculosis (TB), a mycobacterial disease, has deep roots in prehistory, and it remains a major public health concern throughout much of the world today (Bates and Stead, 1993; Daniel, 2006, 2009; Davies et al., 1999; Dheda et al., 2016; Grange et al., 2001; Grange and Zumla, 2002; Maher and Raviglione, 2005; Stone et al., 2009). It is of particular interest to paleopathologists because TB affects bone, so it is detectable in archaeological skeletons. That allows the disease to be identified where medical or historical sources are absent, scarce, or otherwise uninformative (Arriaza et al., 1995; Mays et al., 2001; Nicklisch et al., 2012; Ortner, 2003; Roberts and Buikstra, 2003; Steinbock, 1976).” (page 88 – Introduction section).

“Despite the importance of establishing where and when a disease such as TB was present, documenting its existence does not tell us whether the disease was common or not, hence its impact on past communities. Estimating disease prevalence in archaeological samples, let alone establishing what happened in once-living populations, requires a different approach involving a wide array of skeletal lesions, not just the most distinctive ones, each with an estimated sensitivity and specificity (Boldsen, 2001, 2005a, 2005b, 2008; Boldsen and Mollerup, 2006; Boldsen et al., 2013; Milner and Boldsen, 2017). … Such estimates, which accommodate the diagnostic efficacy of different skeletal lesions, are an important component of paleoepidemiological studies (Boldsen, 2005a; Milner and Boldsen, 2017).” (page 88 – Introduction section).

“The identification of highly distinctive lesions, eliminating alternatives through differential diagnosis, is quite useful when the objective is determining whether a particular disease happened to be present in a specific time and place (Buikstra, 1976). Tallies of skeletons with classic expressions of a disease, however, will do nothing to help estimate disease prevalence in the past, one of the principal objectives of paleoepidemiology (Milner and Boldsen, 2017). Establishing how common that disease happened to be is an essential part of assessing its impact on past communities.” (page 98 – Conclusion section).

The aforementioned two publications have been considered when we stated that palaeopathology focuses on the individual:

“Consequently, it focuses on the individual and its main objective is to determine where and when a disease was present in the past by establishing a definitive (individual) diagnosis, which is based on the recognition of pathological bony changes and their distribution pattern in the skeleton, and elimination of alternative aetiologies through differential diagnosis [4-6].” (lines 54–57 in the originally submitted version). 

“Initially, palaeopathology focuses on the individual, and its main objective is to determine where and when a disease was present in the past [4-7]. Establishment of a definitive (individual) diagnosis is based on the recognition of pathological bony changes and their distribution pattern in the skeleton and elimination of alternative aetiologies through differential diagnosis [4-7].” (lines 54–58 in the revised version)

Based on the above, it would be palaeoepidemiology that concentrates on the population (community) – at least in our understanding.

We agree with Reviewer 1 that the establishment of a definitive diagnosis is not possible in numerous cases, especially if we can rely solely on macroscopic observations. In our manuscript, we have already mentioned some of the factors that can hinder palaeopathologists in arriving at a diagnosis (lines 59–71 in the originally submitted version and lines 58–71 in the revised version). One of them is that most of the macroscopic bony changes we can use as diagnostic criteria – or even their co-occurrence and distribution pattern in the skeleton – cannot be considered specific/pathognomonic to a certain disease. Nonetheless, we still think that the aim of palaeopathology would be the establishment of a definitive diagnosis. We consider differential diagnosis more like a tool, which can help us in achieving a definitive diagnosis by the elimination of alternative aetiologies. We agree with Reviewer 1 that usually, two or more potential aetiologies remain in the differential diagnosis when there is only macroscopic observations – in some cases, further investigations (e.g., microscopic or palaeomicrobiological analyses) can further narrow down the options to the most likely one(s). Fortunately, in the cases presented in our manuscript, the macroscopic observations alone have made it possible to arrive at a diagnosis. This is because in all five presented cases (HK199, HK201, HK225, HK253, and HK309), granular impressions (GIs) were detected on the endocranial surface of the skull, and based on the very recent findings of Spekker and her colleagues (2020 and 2022) (reference nos. 95 and 98 in the originally submitted version and reference nos. 66 and 69 in the revised version), these alterations can be considered as specific/pathognomonic features of TB meningitis. Thus, GIs are sufficient enough on their own to make a definitive diagnosis of the disease in archaeological cases – such as in the five presented cases from the Sarmatian-period archaeological site of Hódmezővásárhely–Kenyere-ér, Bereczki-tanya.

We hope that Reviewer 1 will accept our argumentation.

3) Reviewer 1 noted that “Age at death and biological sex were estimated through different standard methods but, for each individual skeleton, we do not know which methods were used to estimate those parameters. That information must be added to each individual paleodemographic details (page 16 and after, line 232 and after).”. 

Following Reviewer 1’s advice, the methods that have been used to estimate age-at-death and determine sex have been added for each examined individual in the relevant part of the Methods section (lines 179–193 in the revised version):

• “HK199 (object no./stratigraphic no.: 199/227): a younger to middle-aged adult (c. 20–39/49 years old [39,43,45]) female [46-47]. This skeleton is relatively complete and fairly preserved; the remains were unearthed from the first cemetery (Fig 1C); 

• HK201 (object no./stratigraphic no.: 201/229): a middle-aged to older adult (c. 40–x years old [39,43]) individual of indeterminant sex [46]. This skeleton is very incomplete and poorly preserved; the remains were uncovered from the first cemetery (Fig 1C);

• HK225 (object no./stratigraphic no.: 225/302): a juvenile (c. 14–16 years old [40]) individual of indeterminant sex [46-47]. This skeleton is relatively complete and well-preserved; the remains were discovered in a pit from the Sarmatian-period settlement;

• HK253 (object no./stratigraphic no.: 253/358): a younger adult (c. 20–29 years old [38-39,43-45]) female [46-47]. This skeleton is relatively complete and fairly preserved; the remains were unearthed from the second cemetery (Fig 1D); and

• HK309 (object no./stratigraphic no.: 309/362): a middle-aged to older adult (c. 50–x years old [37,39,41,43-45]) male [46-47]. This skeleton is relatively complete and fairly preserved; the remains were uncovered from the second cemetery (Fig 1D).”.

We hope that Reviewer 1 will be satisfied with the adjusted Methods section.

4) Reviewer 1 asked that “Please remove the tag «object» to the individuals, human skeletal remains are not objects and we shouldn’t reify that.”. 

We agree with Reviewer 1 that human skeletons are not objects, it was not our intention at all to reify that. Nonetheless, in our opinion, we have never referred to the skeletons as objects in our manuscript. Where we have mentioned “object nos.” it was because object nos. are part of the ID nos. of the skeletons (lines 232, 235, 238, 241, 244, and 249–251 in the originally submitted version). As we have indicated in the manuscript, ID nos. are composed of an object no. and a stratigraphic no., with a slash symbol between the two (e.g., object no./stratigraphic no.: 199/227 (in line 249 of the originally submitted version) – this is the ID no. of the individual (skeleton) we referred to as HK199). Thus, the tag “object” is not associated with the individuals (skeletons) themselves but with their ID nos. The Hódmezővásárhely–Kenyere-ér, Bereczki-tanya archaeological site contained both settlement features (e.g., houses, storage pits, and trenches) and burials spanning multiple layers and time periods. To properly document findings like these, archaeologists have implemented a system in which object nos. and stratigraphic nos. are used so that the findings can unmistakably be identified and differentiated from each other. Like in many other sites, this system has been applied to the excavation of the Hódmezővásárhely–Kenyere-ér, Bereczki-tanya archaeological site – this is why, instead of using grave nos., the human skeletons have been distinguished by ID nos. that are composed of an object no. and a stratigraphic no., with a slash between the two. In this context, it is unfortunate that in English, the term “object” has several meanings. In Hungarian, we have two terms to use for “object” – 1) “objektum”, which is not pejorative at all and 2) “tárgy”, which, depending on the context, can be pejorative as it can imply objectification of an individual. In the Hungarian archaeological terminology, it is “objektum szám” instead of “tárgy szám”, what we use (‘szám” means “number” in Hungarian). Nonetheless, following Reviewer 1’s comment, we have gone through the text of our manuscript again and found one sentence in which our phrasing could have implied objectification of an individual: “From the Sarmatian-period archaeological site of Hódmezővásárhely–Kenyere-ér, Bereczki-tanya, a total of 28 human skeletons were unearthed – 27 were uncovered from the three cemeteries, whereas one (object no. 225) was discovered in a pit found among the Sarmatian-period settlement remains.” (lines 186–189 in the originally submitted version). Thus, this sentence has been slightly rephrased: “From the Sarmatian-period archaeological site of Hódmezővásárhely–Kenyere-ér, Bereczki-tanya, a total of 28 human skeletons were unearthed – 27 were uncovered from the three cemeteries, and one was discovered in a pit (object no. 225) found among the Sarmatian-period settlement remains.” (lines 132–135 in the revised version). 

We hope that Reviewer 1 will accept our argumentation and agree with our decision to keep the term “object no.” as it is not something we have used to objectify the examined individuals but an archaeological term.

5) Reviewer 1 noted that “Except for the juvenile individual, a TB diagnosis was achieved based solely on intracranial lesions. Even though there are good signs that TB is associated with granular impressions in the inner skull surfaces, there are also competing hypotheses for these lesions that must be handled in a differential diagnosis, especially because no other TB-related lesions were observed.”.

We agree with Reviewer 1 that even if endocranial granular impressions (GIs) have been described as pathognomonic/specific features of TB meningitis by Schultz back in the 1990s (reference nos. 91–93 in the originally submitted version and reference nos. 62–64 in the revised version), more than a decade ago, the diagnostic value of these alterations in the palaeopathological interpretation of TB have been questioned by Roberts and her co-workers (2009) (https://onlinelibrary.wiley.com/doi/10.1002/ajpa.21056). Their argument has been summarised by Spekker et al. (2020) (reference no. 95 in the originally submitted version and reference no. 66 in the revised version):

“Roberts and her colleagues [33] have argued against that GIs can be of tuberculous origin. Their basis of argument is that: 1) TBM is not always accompanied by the formation of meningeal tubercles that could result in pressure atrophy on the inner surface of the skull, and consequently, the development of impressions; 2) in cases with macroscopically visible meningeal tubercles, no endocranial changes were described in the modern medical literature; 3) the typical course of TBM is not long enough to allow bony changes to occur on the inner surface of the skull; and 4) only a weak association between the presence of GIs and TB have been found by Hershkovitz and his colleagues [31] during the examination of skeletons of known cause of death from the Hamann–Todd Collection. (However, their investigations focused not on GIs but on serpens endocrania symmetrica, another endocranial alteration type that may be associated with TBM but not pathognomonic to the disease.) According to Roberts and her colleagues [33], further investigations on GIs in skeletons of known cause of death or in skeletons from osteoarchaeological series with an independent confirmation of the diagnosis of TB (e.g., by light and scanning electron microscopy [27] or by biomolecular methods, such as ancient DNA, lipid biomarker, and extracellular matrix (ECM) protein [42] analyses) are needed to clarify the exact etiology of GIs.” (pages 6 and 8 – Discussion and conclusions section).

Following Roberts and her co-workers’ suggestion, in 2016, a detailed macromorphological analysis has been conducted by Olga Spekker (the first author of the present manuscript) on 427 pre-antibiotic era skeletons with known cause of death from the Robert J. Terry Anatomical Skeletal Collection. One of the aims of this research has been to expand knowledge and understanding about the development of GIs. Furthermore, to improve the palaeopathological interpretation of this endocranial alteration type, along with strengthening its diagnostic value in the identification of TB meningitis in human osteoarchaeological series. The results of this investigation, confirming that GIs are specific/pathognomonic signs of TB meningitis, have been published in Spekker et al. (2020) (reference no. 95 in the originally submitted version and reference no. 66 in the revised version). In this study, the authors have confuted Roberts and her colleagues’ argument (2009):

“Besides the small tubercles formed in the meninges, characteristic pathological features of TBM include enhancing basal meningeal exudate, progressive hydrocephalus, and vasculitis of blood vessels adjacent to or traversing the exudate [34,43,45,52,55-56]; as a rule, the formation of tubercles precedes the development of the latter ones [35].” (page 9 – Discussion and conclusions section).

“If left untreated, TBM usually leads to death within 4–6 weeks after the onset of its symptoms [63]; nonetheless, in some cases [e.g., 64–70], the disease has a protracted course that can last for several months or even years. Therefore, in such cases, the duration of the disease is long enough to allow bony changes to occur on the inner surface of the skull.” (page 9 – Discussion and conclusions section).

“Using modern medical knowledge, paleopathologists endeavor to establish a retrospective diagnosis of TB by macroscopically identifying pathological conditions (e.g., spinal TB and TB arthritis of the large, weight-bearing joints) in human skeletons that may be related to the disease [16-17]. However, utilization of modern diagnostic criteria for TB in the paleopathological practice may not always be appropriate. On the one hand, probable TB-related bony changes observed in recent cases may differ from those detectable in ancient human bone remains (due in part to the introduction of antibiotics in the treatment of the disease) [16,18-20]. On the other hand, in living TB patients, bony changes cannot be surveyed with macromorphological methods but with medical imaging techniques, such as X-ray radiography, computed tomography (CT), and magnetic resonance imaging (MRI), only [18-19]. Nevertheless, subtle bony alterations are mostly impossible to be visualized by modern imaging methods [18-19]. Therefore, subtle bony changes are not relevant to the diagnosis of TB in living patients and are not described as diagnostic criteria for the disease by physicians in the modern medical literature, even if they can be potentially important elements of TB identification for paleopathologists [16,19,21].” (pages 2–3 – Introduction section).

“Although TBM occurs in less than 1% of all cases with active TB [58] and the vast majority of the individuals in our TB group were identified to have died of pulmonary TB (only one of them was recorded to have died of TBM) (S1 Table), about one-third of them revealed GIs suggestive of TBM on the endocranial surface. Our results fit in with those of autopsy studies revealing that a large number of individuals died of pulmonary TB without developing neurological signs and symptoms exhibited tubercles in the CNS. This indicates that involvement of the CNS in pulmonary TB is quite common [93]. Some recent studies showed that about three-fourths of the patients with CNS TB had pulmonary TB 6–12 months prior to the onset of neurological symptoms [61].” (page 11 – Discussion and conclusions section).

“Although no endocranial alterations were described in cases with macroscopically visible meningeal tubercles in the current medical literature, it does not mean that GIs cannot be considered as diagnostic criteria in the paleopathological practice. On the one hand, bony changes associated with TBM were distinctly described in the pathological literature from the first half of the 20th century (preceding the introduction of antibiotics in the management of TB) [e.g., 34-36]. At autopsy of TBM patients, groups of isolated but mostly confluent, small, and dimpled impressions established by pressure atrophy of the tubercles (i.e., GIs), vestiges of hemorrhages developed in close vicinity to the affected blood vessels (i.e., abnormal blood vessel impressions and periosteal appositions), and certain roughnesses indicating characteristic resorption of the bone tissue (i.e., very flat and small erosive bone loss that can be recognized only with low-power microscopy) were observed on the endocranial surface of the skull base and of the lower lateral skull vault after the removal of the dura mater [34-36]. However, autopsy practices have changed over time, and in the present, the dura mater is not completely removed from the basal areas of the skull; thus, there is no detection of the aforementioned bony changes on the endocranial surface. Additionally, the identification of TBM in living patients is usually based on clinical signs and symptoms, CSF findings, and radiological characteristics [43-47,52-53,55]. Similar to periosteal new bone formations on the visceral surface of ribs [18-19], GIs have a very subtle appearance and may be impossible to be visualized by the modern medical imaging techniques. On the other hand, the manifestation of TBM in past human populations may differ from that of modern medical cases due in part to the introduction of antibiotics in the treatment of TB; and therefore, probable TB-related bony changes, including GIs, may not occur in recent cases [16,18-19,21].” (pages 9–10 – Discussion and conclusions section).

In addition, although in the article by Roberts and her colleagues (2009) in which the diagnostic value of GIs has been questioned, the authors have suggested that the lesions, which have been described in a particular case (published by Kappelman et al. in 2008 – https://onlinelibrary.wiley.com/doi/10.1002/ajpa.20739) could have been impressions of the arachnoid granulations instead of GIs, Spekker et al. (2020) (reference no. 95 in the originally submitted version and reference no. 66 in the revised version) have drawn the reader’s attention to the following:

“It should be noted that although having very similar names in the scientific literature, granular impressions and granular foveolae (i.e., impressions of the arachnoid granulations) should not be mistaken for each other, since they refer to two different lesion types affecting the inner surface of the skull.” (page 3 – Introduction section).

We do not want to take sides in this particular case, as it does not matter if it was GIs or granular foveolae, which have been observed by Kappelman et al. (2008). What matters is that it was GIs that have been studied by Spekker and her colleagues (2020) in the Robert J. Terry Anatomical Skeletal Collection, and it was GIs that have been detected in the five cases presented in the current manuscript.

As we have already mentioned above, the findings of Spekker et al. (2020) (reference no. 95 in the originally submitted version and reference no. 66 in the revised version) confirm that GIs are specific/pathognomonic to TB meningitis:

“During the macroscopic evaluation of the 427 selected skeletons with sectioned skulls from the Terry Collection, we found that GIs were ten times more common in individuals recorded to have died of TB than in individuals identified to have died of causes other than TB. Our findings are constituting evidence that there is a positive correlation between GIs and TB. The results of our research project fit in with those of previous studies [e.g., 24–27] concerning the specificity of GIs for TBM, as GIs affected only six individuals in the NTB group (S2 Table). Five out of the above-mentioned six individuals show probable TBM-associated endocranial alterations other than GIs (four cases: abnormally pronounced digital impressions ([e.g., 23-26,71] and one case: periosteal appositions [e.g., 23-26]) (S4 Table) and/or likely TB-related non-endocranial bony changes (two cases: periosteal new bone formations on the visceral surface of ribs [e.g., 16,18-21], two cases: vertebral hypervascularization [e.g., 15,17,32,72-73], two cases: signs of extra-spinal osteomyelitis [e.g., 74-75], and one case: signs of hypertrophic pulmonary osteopathy [e.g., 31,38-40,76]) (S5 Table). It must be noted that even if the recorded cause of death of individuals surveyed in the Terry Collection may not have been TB, individuals could still have suffered from the disease but their death was attributed to another medical condition [18-19]. Moreover, there is always the possibility that an inaccurate cause of death was registered on the morgue record and/or death certificate of individuals from the Terry Collection. Thus, it is possible that in the aforementioned six cases, the observed endocranial and non-endocranial bony changes resulted from TB. In summary, the findings of our study confirm those of Schultz [e.g., 24-26] and Schultz & Schmidt-Schultz [27] that GIs can be considered as pathognomonic features of TBM; and therefore, the paleopathological diagnosis of TBM can be established with a high certainty when GIs are present in ancient human bone remains. The localization pattern and distribution of GIs on the endocranial surface resemble that of the tubercles observed in the affected meninges during the pathogenesis of TBM that further strengthens their tuberculous origin.” (page 10 – Discussion and conclusions section).

The findings of another, very recently published paper by Spekker and her co-workers (2022) (reference no. 98 in the originally submitted version and reference no. 69 in the revised version) have further highlighted the diagnostic value of GIs in the palaeopathological identification of TB meningitis:

“The generated sensitivity and specificity estimate values for the four endocranial alteration types indicative of TBM can be seen in Table 2. Similar to the previously published χ2 test results (Spekker et al., 2020a,b), the calculated sensitivity and specificity estimate values for GIs, ABVIs, and PAs support that there is a positive association between the aforementioned lesion types and TB, as their sensitivity is more than one minus their specificity (Table 2) (Boldsen, 2001; Dangvard Pedersen et al., 2019). It also means that the probability of presenting GIs, ABVIs or PAs is higher in individuals with TB than in those without TB.” (pages 448–449 – Results and discussion section).

“As for the sensitivity estimate values (Table 2), if GIs, ABVIs or PAs are present on the inner skull surface, there is a 29.06%, 21.37% or 20.09% probability of a true TB diagnosis, respectively. On the one hand, the findings (Table 2) indicate that GIs, ABVIs, and PAs cannot be used to identify a large number of TB cases due to the rather high false negative rates (i.e., 0.7094, 0.7863, and 0.7991, respectively); therefore, these lesion types are not suitable for screening purposes. In contrast to GIs, ABVIs, and PAs, the sensitivity of APDIs is rather high (i.e., 0.6581); meaning that with the application of APDIs as diagnostic criteria, quite a large number of individuals with TB can be recognized (Spekker et al., 2021). Nevertheless, it should not be expected that any of the four examined endocranial alteration types could be used to identify all individuals with TB, as they can develop only in case of meningeal involvement (i.e., TBM) that is a very rare form of TB, occurring in less than 1% of the active TB cases today (Myers, 2007).” (page 449 – Results and discussion section).

“On the other hand, the results (Table 2) suggest that GIs can be considered as specific signs of TB due to the extremely low false positive rate (i.e., 0.0311); consequently, GIs are sufficient enough on their own to make a definitive diagnosis of TBM. It is not surprising if we consider that GIs are defined as endocranial impressions established by pressure atrophy of the tubercles (formed on the outermost meningeal layer) that are the hallmark features of TB (Schultz, 1993, 1999, 2001, 2003; Schultz & Schmidt-Schultz, 2015; Spekker et al., 2020a). Although in case of ABVIs and PAs (Table 2), the generated false positive rates are quite low (i.e., 0.0622 and 0.1036, respectively), they are not sufficient enough on their own to make a definitive diagnosis of TBM, since besides TBM, other medical conditions can also result in the development of ABVIs and PAs (Schultz, 1993, 1999, 2001, 2003; Schultz & Schmidt-Schultz, 2015; Spekker et al., 2020b). It should be mentioned that similar to ABVIs and PAs, APDIs are not specific to TBM (Table 2) – among the four evaluated endocranial alteration types, APDIs had the lowest specificity estimate value (i.e., 0.6788) (Spekker et al., 2021). Assumed from the above, the paleopathological diagnosis of TBM cannot be established based only on the presence of ABVIs, PAs or APDIs on the inner skull surface.” (pages 449–450 – Results and discussion section).

“The χ2 test results regarding the association of ABVIs, PAs, and APDIs (Table 3) revealed that their co-occurrence with each other (in any possible combination) is significantly more common in individuals who died of TB than in those who died of non-TB causes. This implies that in cases where these lesions concomitantly occur with each other, the chance of them being tuberculous in origin is higher than in cases where they are present alone. Although the co-occurrence of ABVIs, PAs, and/or APDIs on the endocranial surface without the simultaneous presence of GIs is still not enough to make a definitive diagnosis of TBM (as even the co-occurrence of ABVIs, PAs, and APDIs (in any possible combination) can result from pathological conditions other than TBM), if ABVIs, PAs, and/or APDIs are concurrently present with GIs on the inner skull surface, their tuberculous origin is very likely.” (page 450 – Results and discussion section).

Based on the above, we consider GIs as specific/pathognomonic features of TB meningitis, and this is why, at least in our opinion, there is no need for a differential diagnosis (as TB meningitis is the only aetiology that should be considered in it): 

“Recently, it has been confirmed that GIs can be considered as specific signs of TB meningitis; therefore, they are sufficient enough on their own to make the definitive diagnosis of the disease in HK199, HK201, HK225, HK253, and HK309 [91,94-95,98].” (lines 427–430 in the originally submitted version).

“It has been recently confirmed that GIs can be considered as specific signs of TB meningitis; and therefore, are sufficient enough on their own to make a definitive diagnosis of the disease in HK199, HK201, HK225, HK253, and HK309 [62,65-66,69].” (lines 375–378 in the revised version).

We hope that Reviewer 1 will accept our argumentation.

6) Reviewer 2 commented that “Olga Spekker and colleagues subjected in their study “White Plague among the “Forgotten People” from the Barbaricum of the Carpathian Basin – Cases with tuberculosis from the Sarmatian-period (3rd–4th centuries CE) archaeological site of Hódmezővásárhely–Kenyere-ér, Bereczki-tanya (Hungary) 28 skeletal individuals from the Sarmatian period (1st–5th centuries CE) in the Barbaricum of the Carpathian Basin to a detailed paleopathological evaluation. Thereby they identified 5 individuals that display bone changes indicative for Tuberculosis meningitis. One of the five TB cases displayed in addition skeletal signs for osteoarticular TB. Overall, the methods applied in this study are well described and the obtained results are interesting for the fields of anthropology, paleopathology, archaeology, historical research, and paleomicrobiology.”

We are very grateful for Reviewer 2 for their positive feedback on our research. We appreciate the points they have raised and we have responded to each below.

7) Reviewer 2 commented that “I would like to ask the authors to shorten the introduction of the Sarmatian-period in the introductory part (line 85 to 150). This very detailed description distracts in my opinion a bit from the actual topic of the paper.”. 

We agree with Reviewer 2 that the very detailed description of the Sarmatian period we have provided in the Introduction section can distract the reader’s attention. On the other hand, we think that this detailed description can be useful if someone is not familiar with the Sarmatian period of the Carpathian Basin. Thus, we have decided to move most of the detailed description of the Sarmatian period (lines 85–134 in the originally submitted version) – along with the references that have been cited in this part of the manuscript (reference nos. 22–51 in the originally submitted version) – to the supplementary material as supplementary text 1 (S1 Text); consequently, the subsequent references have been renumbered. In our opinion, this way we can make sure that this detailed description will not distract the reader’s attention, but its full content will still be available for them. We have kept some sentences (lines 135–150 and 85–98 in the originally submitted and revised versions, respectively) in the main text – some of them contain important information about the possible explanation of why the number of reported TB cases from the Sarmatian-period Barbaricum of the Carpathian Basin is low, and others contain information about the aims of our study.

We hope that Reviewer 2 will be satisfied with the adjusted Introduction section.

8) Reviewer 2 noted that “Beside one minor comment I would like to ask the authors to comment in the manuscript in more detail on the possibility that the observed skeletal changes can derive from infectious/metabolic disease others than TB. I assumed so far that there exists only one clear cut clinical picture in skeletal remains for TB, the Potts disease. All other skeletal changes linked to TB (e.g. on ribs or the skull) are indicative for this disease but it cannot be excluded that these changes are caused by other diseases. I am not a paleopathologist so please correct me if I am wrong. I saw that you also talk in your paleopathological methods part of skeletal alterations that are “indicative of TB” (line 220, 223) or “suggestive of TB meningitis” (line 226). If these are first indications of TB in these individuals than I would like to ask you to also consider this throughout your manuscript by: Toning down the title “Presence of tuberculosis in the Sarmatian-period…” in the discussion, And by including a paragraph in the discussion highlighting the possibility that these changes could potentially also be caused by other diseases.”.

We thank Reviewer 2 for noting this important point. We have two reasons why we have used “indicative of” and “suggestive of” when we have discussed about the bony changes that have been associated with different forms of TB in the palaeopathological literature and have been used as diagnostic criteria in the present study. On the one hand, the vast majority of these skeletal lesions (including the ones that can be related to spinal TB (also known as Pott’s disease), which is a form of osteoarticular TB) are not specific/pathognomonic to TB (some of them, such as the Pott’s gibbus that can develop in later stages of spinal TB, are more characteristic than the others). The only exception is the endocranial granular impressions (GIs), which seem to be specific/pathognomonic to TB meningitis, based on the very recent findings of Spekker et al. (2020 and 2022) (reference nos. 95 and 98 in the originally submitted version and reference nos. 66 and 69 in the revised version). On the other hand, except for osteoarticular TB, which directly affects the skeleton, other forms of TB (e.g., pulmonary TB/TB pleurisy or TB meningitis) can only indirectly affect the skeleton and consequently, leave bony changes. Thus, in these instances, the skeletal lesions are only the signs, being specific or not specific, of these extra-skeletal TB forms – it is not the extra-skeletal TB form that can directly be examined but only its (indirect) signs (in the form of bony changes).

We hope that Reviewer 2 will accept our argumentation.

As we have already mentioned above, we consider GIs as specific/pathognomonic features of TB meningitis, and this is why, at least in our opinion, there is no need for a differential diagnosis (as TB meningitis is the only aetiology that should be considered in it): 

 “Recently, it has been confirmed that GIs can be considered as specific signs of TB meningitis; therefore, they are sufficient enough on their own to make the definitive diagnosis of the disease in HK199, HK201, HK225, HK253, and HK309 [91,94-95,98].” (lines 427–430 in the originally submitted version).

“It has been recently confirmed that GIs can be considered as specific signs of TB meningitis; and therefore, are sufficient enough on their own to make a definitive diagnosis of the disease in HK199, HK201, HK225, HK253, and HK309 [62,65-66,69].” (lines 375–378 in the revised version).

We hope that Reviewer 2 will accept our argumentation.

As for the other three endocranial alteration types (abnormal blood vessel impressions – ABVIs, periosteal appositions – PAs, and abnormally pronounced digital impressions – APDIs), based on the findings of Spekker et al. (2022) (reference no. 98 in the originally submitted version and reference no. 69 in the revised version), although these lesions are not specific/pathognomonic to TB meningitis, they can be of tuberculous origin, especially if they simultaneously occur with each other and/or GIs:

“The χ2 test results regarding the association of ABVIs, PAs, and APDIs (Table 3) revealed that their co-occurrence with each other (in any possible combination) is significantly more common in individuals who died of TB than in those who died of non-TB causes. This implies that in cases where these lesions concomitantly occur with each other, the chance of them being tuberculous in origin is higher than in cases where they are present alone. Although the co-occurrence of ABVIs, PAs, and/or APDIs on the endocranial surface without the simultaneous presence of GIs is still not enough to make a definitive diagnosis of TBM (as even the co-occurrence of ABVIs, PAs, and APDIs (in any possible combination) can result from pathological conditions other than TBM), if ABVIs, PAs, and/or APDIs are concurrently present with GIs on the inner skull surface, their tuberculous origin is very likely.” (page 450 – Results and discussion section).

In the originally submitted version of our manuscript (lines 464–469) we have already tried to highlight that ABVIs, PAs, and APDIs are not specific to TB meningitis; and thus, other medical conditions have to be considered in their differential diagnosis, and it is their co-occurrence with GIs in HK199 and HK225 what makes their tuberculous origin very likely:

“It is important to note that in contrast to GIs, ABVIs, PAs, and APDIs are not pathognomonic for TB meningitis, since they can be generated by many other infectious and non-infectious conditions (e.g., non-specific and specific meningitis, traumas, haemorrhages, brain tumours, and/or scurvy) [90,92-93,100,112,122,124,140-148]. Nevertheless, their concurrent presence with GIs in HK199 and HK225 implies that the detected ABVIs, PAs, and APDIs could have most likely resulted from TB meningitis [98].”

To make our argument clearer for the reader, we slightly rephrased the aforementioned text in the revised version of our manuscript (lines 413–421):

“It is important to note that in contrast to GIs, the other three endocranial alteration types (ABVIs, PAs, and APDIs) observed in HK199 and/or HK225 are not pathognomonic for TB meningitis – they can be generated by many other infectious and non-infectious conditions. Besides TB, numerous other aetiologies should also be considered in the differential diagnosis of ABVIs, PAs, and APDIs. They can include: non-specific and specific meningitis, traumas, haemorrhages, brain tumours, and/or scurvy [61,63-64,71,83,93,95,111-119]. Nevertheless, the concurrent presence of these endocranial alterations along with GIs in HK199 and HK225 indicates that the detected ABVIs, PAs, and APDIs are most likely resultant from TB meningitis [69].”

We hope that Reviewer 2 will accept our argumentation and will be satisfied with the adjusted text.

Following Reviewer 2’ advice, the Discussion section has been adjusted – the differential diagnoses of the observed non-endocranial bony changes have been added (some references (reference nos. 142–145 and 152–156 in the revised version) have also been added and the subsequent references have been renumbered) and two supplementary texts (S2 Text and S3 Text), containing detailed information about these differential diagnoses, have been created and added:

“Numerous differential diagnoses, such as pyogenic spinal infections and granulomatous spinal infections other than TB (e.g., brucellosis or aspergillosis) (S2 Text) [122,142-145], need to be considered for the bony changes observed in the vertebral column of HK225. Based on their nature, association, and distribution pattern, they are most consistent with spinal TB. Their concurrent presence with endocranial alterations indicative of TB meningitis further supports their tuberculous origin in HK225.” (lines 466–471 in the revised version)

“Based on the concurrent presence of localised peri-articular osteoporosis along with endocranial alterations indicative of TB meningitis and with vertebral lesions indicative of spinal TB, TB arthritis seems to be its most likely underlying cause in HK225. Other aetiologies could still be considered in the differential diagnosis, such as rheumatoid arthritis and pyogenic arthritis (S3 Text) [152-156].” (lines 481–485 in the revised version)

“S2 Text. Differential diagnoses of the skeletal lesions indicative of tuberculous involvement of the spine that were detected in HK225.”

“S3 Text. Differential diagnoses of the bony changes indicative of tuberculous involvement of both hip joints that were observed in HK225.”

We hope that Reviewer 2 will accept our argumentation, and will be satisfied with the adjusted text and the created supplementary texts (S2 Text and S3 Text).

Responses to the Academic Editor’s comments:

1) “Please double check that all references are correctly formatted according to the style of Plos one.”.

We have checked again the references and made some minor changes to make sure that their format follows the requirements of PLOS ONE. All changes made have been highlighted in the revised version with track changes.

2) “Please add to the ethics statement some consideration regarding the fact that the remains were treated with dignity and respect during investigation, and consider adding, as a reference, the ethical recommendation made by Squires et al. released in 2022 on the AJBA.”.

We thank the Academic Editor for noting this important point. Following the suggestion, a sentence has been included into our Ethics Statement: “The research has been conducted in an ethically responsible manner – the bone remains of HK199, HK201, HK225, HK253, and HK309 have been examined with dignity and respect.” (lines 207–208 in the revised version).

3) “Please use just either British or American spelling but not a mixture of the two.” and “Please have the manuscript read by a native English speaker prior to re-submission, which is going to help for clarity.”.

The manuscript has been copyedited for language usage, spelling, and grammar by Dr David R. Hunt, PhD, a forensic anthropologist and the former curator of the Robert J. Terry Anatomical Skeletal Collection (Washington, DC, USA), who is a native English speaker. Dr Hunt has been included as an author in the revised version of the manuscript.

In the revised version of our manuscript, we have tried to execute all suggestions of the reviewers. We hope this new version will be suitable for publication in PLOS ONE.

Thank you again for the reviewers’ insightful and constructive comments and your editorial work!

Yours sincerely,

Dr Olga Spekker, PhD

Corresponding author

---

## [Editor Report · Decision Letter 1]

2 Nov 2023

PONE-D-23-20370R1White Plague among the “Forgotten People” from the Barbaricum of the Carpathian Basin – Cases with tuberculosis from the Sarmatian-period (3rd–4th centuries CE) archaeological site of Hódmezővásárhely–Kenyere-ér, Bereczki-tanya (Hungary)PLOS ONE

Dear Dr. Spekker,

Thank you for submitting your manuscript to PLOS ONE. After careful consideration, we feel that it has merit but does not fully meet PLOS ONE’s publication criteria as it currently stands. Therefore, we invite you to submit a revised version of the manuscript that addresses the points raised during the review process.

We look forward to receiving your revised manuscript.

Kind regards,

Dario Piombino-Mascali, Ph.D.

Academic Editor

PLOS ONE

Journal Requirements:

Additional Editor Comments:

Dear Dr Spekker, I have emailed you requesting some minor changes. Also, if you can, please use your institutional email address, rather than gmail, for paper correspondence. 

---

## [Author Response · Author response to Decision Letter 1]

6 Nov 2023

Dr Dario Piombino-Mascali, PhD

Academic Editor

PLOS ONE

Dear Dr Piombino-Mascali,

We are very thankful for your comments regarding our manuscript entitled “White Plague among the “Forgotten People” from the Barbaricum of the Carpathian Basin – Cases with tuberculosis from the Sarmatian-period (3rd–4th centuries CE) archaeological site of Hódmezővásárhely–Kenyere-ér, Bereczki-tanya (Hungary)” that was submitted to PLOS ONE (manuscript ID: PONE-D-23-20370). The main text has been modified following your suggestions and was copyedited for language usage, spelling, and grammar by Dr David R. Hunt, PhD.

The revised manuscript file has been uploaded to the submission site of the journal. We hope this new version will be suitable for publication in PLOS ONE.

Thank you again for your comments and your editorial work!

Yours sincerely,

Dr Olga Spekker, PhD

Corresponding author

University of Szeged, Közép fasor 52, H-6726, Szeged, Hungary

Email: olga.spekker@gmail.comTel: +36 20 807 72 94

---

## [Editor Report · Decision Letter 2]

8 Nov 2023

White Plague among the “Forgotten People” from the Barbaricum of the Carpathian Basin – Cases with tuberculosis from the Sarmatian-period (3rd–4th centuries CE) archaeological site of Hódmezővásárhely–Kenyere-ér, Bereczki-tanya (Hungary)

PONE-D-23-20370R2

Dear Dr. Spekker,

We’re pleased to inform you that your manuscript has been judged scientifically suitable for publication and will be formally accepted for publication once it meets all outstanding technical requirements.

Kind regards,

Dario Piombino-Mascali, Ph.D.

Academic Editor

PLOS ONE
---

## [Editor Report · Acceptance letter]

14 Dec 2023

PONE-D-23-20370R2 

PLOS ONE

Dear Dr. Spekker, 

I'm pleased to inform you that your manuscript has been deemed suitable for publication in PLOS ONE. Congratulations! Your manuscript is now being handed over to our production team.

Kind regards, 

on behalf of

Dr. Dario Piombino-Mascali 

Academic Editor

PLOS ONE